# AdANNS: A Framework for Adaptive Semantic Search

**Aniket Rege**[*†]   **Aditya Kusupati**[*†◇]   **Sharan Ranjit S**[†]   **Alan Fan**[†]   **Qingqing Cao**[†],
**Sham Kakade**[‡]   **Prateek Jain**[◇]   **Ali Farhadi**[†]
[†]University of Washington, [◇]Google Research, [‡]Harvard University
{kusupati,ali}@cs.washington.edu, prajain@google.com

## Abstract

Web-scale search systems learn an encoder to embed a given query which is then hooked into an approximate nearest neighbor search (ANNS) pipeline to retrieve similar data points. To accurately capture tail queries and data points, learned representations typically are *rigid, high-dimensional* vectors that are generally used as-is in the entire ANNS pipeline and can lead to computationally expensive retrieval. In this paper, we argue that instead of rigid representations, different stages of ANNS can leverage *adaptive representations* of varying capacities to achieve significantly better accuracy-compute trade-offs, i.e., stages of ANNS that can get away with more approximate computation should use a lower-capacity representation of the same data point. To this end, we introduce AdANNS 🎎, a novel ANNS design framework that explicitly leverages the flexibility of Matryoshka Representations [31]. We demonstrate state-of-the-art accuracy-compute trade-offs using novel AdANNS-based key ANNS building blocks like search data structures (AdANNS-IVF) and quantization (AdANNS-OPQ). For example on ImageNet retrieval, AdANNS-IVF is up to $1.5\%$ more accurate than the rigid representations-based IVF [48] at the same compute budget; and matches accuracy while being up to $90\times$ faster in *wall-clock time*. For Natural Questions, 32-byte AdANNS-OPQ matches the accuracy of the 64-byte OPQ baseline [13] constructed using rigid representations – *same accuracy at half the cost!* We further show that the gains from AdANNS translate to modern-day composite ANNS indices that combine search structures and quantization. Finally, we demonstrate that AdANNS can enable inference-time adaptivity for compute-aware search on ANNS indices built non-adaptively on matryoshka representations. Code is open-sourced at https://github.com/RAIVNLab/AdANNS.

## 1   Introduction

Semantic search [24] on learned representations [40, 41, 50] is a major component in retrieval pipelines [4, 9]. In its simplest form, semantic search methods learn a neural network to embed queries as well as a large number ($N$) of data points in a $d$-dimensional vector space. For a given query, the nearest (in embedding space) point is retrieved using either an exact search or using approximate nearest neighbor search (ANNS) [21] which is now indispensable for real-time large-scale retrieval.

Existing semantic search methods learn fixed or *rigid* representations (RRs) which are used as is in all the stages of ANNS (data structures for data pruning and quantization for cheaper distance computation; see Section 2). That is, while ANNS indices allow a variety of parameters for searching the design space to optimize the accuracy-compute trade-off, the provided data dimensionality is typically assumed to be an *immutable* parameter. To make it concrete, let us consider inverted file index (IVF) [48], a popular web-scale ANNS technique [16]. IVF has two stages (Section 3) during inference: (a) *cluster mapping*: mapping the query to a cluster of data points [36], and (b) *linear*

---

[*]Equal contribution.

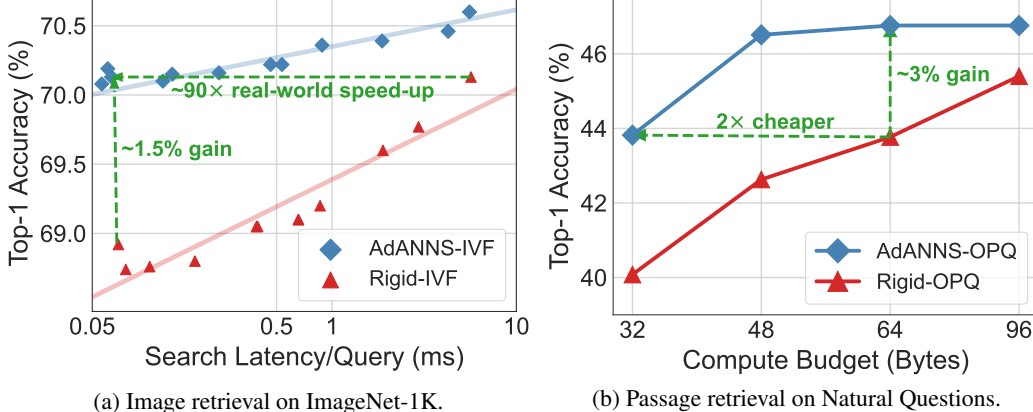

(a) Image retrieval on ImageNet-1K.

(b) Passage retrieval on Natural Questions.

Figure 1: AdANNS helps design search data structures and quantization methods with *better accuracy-compute trade-offs* than the existing solutions. In particular, (a) AdANNS-IVF improves on standard IVF by up to $1.5\%$ in accuracy while being $90\times$ faster in deployment and (b) AdANNS-OPQ is as accurate as the baseline at *half the cost!* Rigid-IVF and Rigid-OPQ are standard techniques that are built on rigid representations (RRs) while AdANNS uses matryoshka representations (MRs) [31].

*scan*: distance computation w.r.t all points in the retrieved cluster to find the nearest neighbor (NN). Standard IVF utilizes the same high-dimensional RR for both phases, which can be sub-optimal.

**Why the sub-optimality?** Imagine one needs to partition a dataset into $k$ clusters for IVF and the dimensionality of the data is $d$ – IVF uses full $d$ representation to partition into $k$ clusters. However, suppose we have an alternate approach that somehow projects the data in $d/2$ dimensions and learns $2k$ clusters. Note that the storage and computation to find the nearest cluster remains the same in both cases, i.e., when we have $k$ clusters of $d$ dimensions or $2k$ clusters of $d/2$ dimensions. $2k$ clusters can provide significantly more refined partitioning, but the distances computed between queries and clusters could be significantly more inaccurate after projection to $d/2$ dimensions.

So, if we can find a mechanism to obtain a $d/2$-dimensional representation of points that can accurately approximate the topology/distances of $d$-dimensional representation, then we can potentially build significantly better ANNS structure that utilizes different capacity representations for the cluster mapping and linear scan phases of IVF. But how do we find such *adaptive representations*? These desired adaptive representations should be cheap to obtain and still ensure distance preservation across dimensionality. Post-hoc dimensionality reduction techniques like SVD [14] and random projections [25] on high-dimensional RRs are potential candidates, but our experiments indicate that in practice they are highly inaccurate and do not preserve distances well enough (Figure 2).

Instead, we identify that the recently proposed Matryoshka Representations (MRs) [31] satisfy the specifications for adaptive representations. Matryoshka representations pack information in a hierarchical nested manner, i.e., the first $m$-dimensions of the $d$-dimensional MR form an accurate low-dimensional representation while being aware of the information in the higher dimensions. This allows us to deploy MRs in two major and novel ways as part of ANNS: (a) low-dimensional representations for accuracy-compute optimal clustering and quantization, and (b) high-dimensional representations for precise re-ranking when feasible.

To this effort, we introduce AdANNS 🏃, a novel design framework for semantic search that uses matryoshka representation-based *adaptive representations* across different stages of ANNS to ensure significantly better accuracy-compute trade-off than the state-of-the-art baselines.

Typical ANNS systems have two key components: (a) search data structure to store datapoints, (b) distance computation to map a given query to points in the data structure. Through AdANNS, we address both these components and significantly improve their performance. In particular, we first propose AdANNS-IVF (Section 4.1) which tackles the first component of ANNS systems. AdANNS-IVF uses standard full-precision computations but uses adaptive representations for different IVF stages. On ImageNet 1-NN image retrieval (Figure 1a), AdANNS-IVF is up to $1.5\%$ more accurate for the compute budget and $90\times$ cheaper in deployment for the same accuracy as IVF.

We then propose AdANNS-OPQ (Section 4.2) which addresses the second component by using AdANNS-based quantization (OPQ [13]) – here we use exhaustive search overall points. AdANNS-OPQ is as accurate as the baseline OPQ on RRs while being at least $2\times$ faster on Natural Questions [32] 1-NN passage retrieval (Figure 1b). Finally, we combine the two techniques to obtain AdANNS-IVFOPQ (Section 4.3) which is more accurate while being much cheaper – up to $8\times$ – than the traditional IVFOPQ [24] index. To demonstrate generality of our technique, we adapt AdANNS to DiskANN [22] which provides interesting accuracy-compute tradeoff; see Table 1.

While MR already has multi-granular representations, careful integration with ANNS building blocks is critical to obtain a practical method and is *our main contribution*. In fact, Kusupati et al. [31] proposed a simple adaptive retrieval setup that uses smaller-dimensional MR for shortlisting in retrieval followed by precise re-ranking with a higher-dimensional MR. Such techniques, unfortunately, cannot be scaled to industrial systems as they require forming a new index for every shortlisting provided by low-dimensional MR. Ensuring that the method aligns well with the modern-day ANNS pipelines is important as they already have mechanisms to handle real-world constraints like load-balancing [16] and random access from disk [22]. So, AdANNS is a step towards making the abstraction of adaptive search and retrieval feasible at the web-scale.

Through extensive experimentation, we also show that AdANNS generalizes across search data structures, distance approximations, modalities (text & image), and encoders (CNNs & Transformers) while still translating the theoretical gains to latency reductions in deployment. While we have mainly focused on IVF and OPQ-based ANNS in this work, AdANNS also blends well with other ANNS pipelines. We also show that AdANNS can enable compute-aware elastic search on prebuilt indices without making any modifications (Section 5.1); note that this is in contrast to AdANNS-IVF that builds the index explicitly utilizing "adaptivity" in representations. Finally, we provide an extensive analysis on the alignment of matryoshka representation for better semantic search (Section 5.2).

**We make the following key contributions:**

- We introduce AdANNS 🏃, a novel framework for semantic search that leverages matryoshka representations for designing ANNS systems with better accuracy-compute trade-offs.
- AdANNS powered search data structure (AdANNS-IVF) and quantization (AdANNS-OPQ) show a significant improvement in accuracy-compute tradeoff compared to existing solutions.
- AdANNS generalizes to modern-day composite ANNS indices and can also enable compute-aware elastic search during inference with no modifications.

## 2   Related Work

Approximate nearest neighbour search (ANNS) is a paradigm to come as close as possible [7] to retrieving the "true" nearest neighbor (NN) without the exorbitant search costs associated with exhaustive search [21, 52]. The "approximate" nature comes from data pruning as well as the cheaper distance computation that enable real-time web-scale search. In its naive form, NN-search has a complexity of $\mathcal{O}(dN)$; $d$ is the data dimensionality used for distance computation and $N$ is the size of the database. ANNS employs each of these approximations to reduce the linear dependence on the dimensionality (cheaper distance computation) and data points visited during search (data pruning).

**Cheaper distance computation.** From a bird's eye view, cheaper distance computation is always obtained through dimensionality reduction (quantization included). PCA and SVD [14, 26] can reduce dimensionality and preserve distances only to a limited extent without sacrificing accuracy. On the other hand, quantization-based techniques [6, 15] like (optimized) product quantization ((O)PQ) [13, 23] have proved extremely crucial for relatively accurate yet cheap distance computation and simultaneously reduce the memory overhead significantly. Another naive solution is to independently train the representation function with varying low-dimensional information bottlenecks [31] which is rarely used due to the costs of maintaining multiple models and databases.

**Data pruning.** Enabled by various data structures, data pruning reduces the number of data points visited as part of the search. This is often achieved through hashing [8, 46], trees [3, 12, 16, 48] and graphs [22, 38]. More recently there have been efforts towards end-to-end learning of the search data structures [17, 29, 30]. However, web-scale ANNS indices are often constructed on rigid $d$-dimensional real vectors using the aforementioned data structures that assist with the real-time search. For a more comprehensive review of ANNS structures please refer to [5, 34, 51].

**Composite indices.** ANNS pipelines often benefit from the complementary nature of various building blocks [24, 42]. In practice, often the data structures (coarse-quantizer) like IVF [48] and HNSW [37] are combined with cheaper distance alternatives like PQ [23] (fine-quantizer) for massive speed-ups in web-scale search. While the data structures are built on $d$-dimensional real vectors, past works consistently show that PQ can be safely used for distance computation during search time. As evident in modern web-scale ANNS systems like DiskANN [22], the data structures are built on $d$-dimensional real vectors but work with PQ vectors $(32 - 64$-byte) for fast distance computations.

**ANNS benchmark datasets.** Despite the Herculean advances in representation learning [19, 42], ANNS progress is often only benchmarked on fixed representation vectors provided for about a dozen million to billion scale datasets [1, 47] with limited access to the raw data. This resulted in the improvement of algorithmic design for rigid representations (RRs) that are often not specifically designed for search. All the existing ANNS methods work with the assumption of using the provided $d$-dimensional representation which might not be Pareto-optimal for the accuracy-compute trade-off in the first place. Note that the lack of raw-image and text-based benchmarks led us to using ImageNet-1K [45] (1.3M images, 50K queries) and Natural Questions [32] (21M passages, 3.6K queries) for experimentation. While not billion-scale, the results observed on ImageNet often translate to real-world progress [28], and Natural Questions is one of the largest question answering datasets benchmarked for dense passage retrieval [27], making our results generalizable and widely applicable.

In this paper, we investigate the utility of adaptive representations – embeddings of different dimensionalities having similar semantic information – in improving the design of ANNS algorithms. This helps in transitioning out of restricted construction and inference on rigid representations for ANNS. To this end, we extensively use Matryoshka Representations (MRs) [31] which have desired adaptive properties in-built. To the best of our knowledge, this is the first work that improves accuracy-compute trade-off in ANNS by leveraging adaptive representations on different phases of construction and inference for ANNS data structures.

## 3 Problem Setup, Notation, and Preliminaries

The problem setup of approximate nearest neighbor search (ANNS) [21] consists of a database of $N$ data points, $[x_1, x_2, \ldots, x_N]$, and a query, $q$, where the goal is to "approximately" retrieve the nearest data point to the query. Both the database and query are embedded to $\mathbb{R}^d$ using a representation function $\phi : \mathcal{X} \rightarrow \mathbb{R}^d$, often a neural network that can be learned through various representation learning paradigms [2, 19, 20, 40, 42].

**Matryoshka Representations (MRs).** The $d$-dimensional representations from $\phi$ can have a nested structure like Matryoshka Representations (MRs) [31] in-built – $\phi^{\mathrm{MR}(d)}$. Matryoshka Representation Learning (MRL) learns these nested representations with a simple strategy of optimizing the same training objective at varying dimensionalities. These granularities are ordered such that the lowest representation size forms a prefix for the higher-dimensional representations. So, high-dimensional MR inherently contains low-dimensional representations of varying granularities that can be accessed for free – first $m$-dimensions ($m \in [d]$) ie., $\phi^{\mathrm{MR}(d)}[1 : m]$ from the $d$-dimensional MR form an $m$-dimensional representation which is as accurate as its independently trained rigid representation (RR) counterpart – $\phi^{\mathrm{RR}(m)}$. Training an encoder with MRL does not involve any overhead or hyperparameter tuning and works seamlessly across modalities, training objectives, and architectures.

**Inverted File Index (IVF).** IVF [48] is an ANNS data structure used in web-scale search systems [16] owing to its simplicity, minimal compute overhead, and high accuracy. IVF construction involves clustering (coarse quantization through k-means) [36] on $d$-dimensional representation that results in an inverted file list [53] of all the data points in each cluster. During search, $d$-dimensional query representation is assigned to the most relevant cluster ($C_i; i \in [k]$) by finding the closest centroid ($\mu_i$) using an appropriate distance metric ($L_2$ or cosine). This is followed by an exhaustive linear search across all data points in the cluster which gives the closest NN (see Figure 5 in Appendix A for IVF overview). Lastly, IVF can scale to web-scale by utilizing a hierarchical IVF structure within each cluster [16]. Table 2 in Appendix A describes the retrieval formula for multiple variants of IVF.

**Optimized Product Quantization (OPQ).** Product Quantization (PQ) [23] works by splitting a $d$-dimensional real vector into $m$ sub-vectors and quantizing each sub-vector with an independent $2^b$

length codebook across the database. After PQ, each $d$-dimensional vector can be represented by a compact $m \times b$ bit vector; we make each vector $m$ bytes long by fixing $b = 8$. During search time, distance computation between the query vector and PQ database is extremely efficient with only $m$ codebook lookups. The generality of PQ encompasses scalar/vector quantization [15, 36] as special cases. However, PQ can be further improved by rotating the $d$-dimensional space appropriately to maximize distance preservation after PQ. Optimized Product Quantization (OPQ) [13] achieves this by learning an orthonormal projection matrix $R$ that rotates the $d$-dimensional space to be more amenable to PQ. OPQ shows consistent gains over PQ across a variety of ANNS tasks and has become the default choice in standard composite indices [22, 24].

**Datasets.**    We evaluate the ANNS algorithms while changing the representations used for the search thus making it impossible to evaluate on the usual benchmarks [1]. Hence we experiment with two public datasets: (a) ImageNet-1K [45] dataset on the task of image retrieval – where the goal is to retrieve images from a database (1.3M image train set) belonging to the same class as the query image (50K image validation set) and (b) Natural Questions (NQ) [32] dataset on the task of question answering through dense passage retrieval – where the goal is to retrieve the relevant passage from a database (21M Wikipedia passages) for a query (3.6K questions).

**Metrics**    Performance of ANNS is often measured using recall score [22], $k$-recall@$N$ – recall of the exact NN across search complexities which denotes the recall of $k$ "true" NN when $N$ data points are retrieved. However, the presence of labels allows us to compute 1-NN (top-1) accuracy. Top-1 accuracy is a harder and more fine-grained metric that correlates well with typical retrieval metrics like recall and mean average precision (mAP@$k$). Even though we report top-1 accuracy by default during experimentation, we discuss other metrics in Appendix C. Finally, we measure the compute overhead of ANNS using MFLOPS/query and also provide wall-clock times (see Appendix B.1).

**Encoders.**    For ImageNet, we encode both the database and query set using a ResNet50 ($\phi_I$) [19] trained on ImageNet-1K. For NQ, we encode both the passages in the database and the questions in the query set using a BERT-Base ($\phi_N$) [10] model fine-tuned on NQ for dense passage retrieval [27].

We use the trained ResNet50 models with varying representation sizes ($d = [8, 16, \ldots, 2048]$; default being 2048) as suggested by Kusupati et al. [31] alongside the MRL-ResNet50 models trained with MRL for the same dimensionalities. The RR and MR models are trained to ensure the supervised one-vs-all classification accuracy across all data dimensionalities is nearly the same – 1-NN accuracy of 2048-$d$ RR and MR models are $71.19\%$ and $70.97\%$ respectively on ImageNet-1K. Independently trained models, $\phi_I^{\mathrm{RR}(d)}$, output $d = [8, 16 \ldots, 2048]$ dimensional RRs while a single MRL-ResNet50 model, $\phi_I^{\mathrm{MR}(d)}$, outputs a $d = 2048$-dimensional MR that contains all the 9 granularities.

We also train BERT-Base models in a similar vein as the aforementioned ResNet50 models. The key difference is that we take a pre-trained BERT-Base model and fine-tune on NQ as suggested by Karpukhin et al. [27] with varying (5) representation sizes (bottlenecks) ($d = [48, 96, \ldots, 768]$; default being 768) to obtain $\phi_N^{\mathrm{RR}(d)}$ that creates RRs for the NQ dataset. To get the MRL-BERT-Base model, we fine-tune a pre-trained BERT-Base encoder on the NQ train dataset using the MRL objective with the same granularities as RRs to obtain $\phi_N^{\mathrm{MR}(d)}$ which contains all five granularities. Akin to ResNet50 models, the RR and MR BERT-Base models on NQ are built to have similar 1-NN accuracy for 768-$d$ of $52.2\%$ and $51.5\%$ respectively. More implementation details can be found in Appendix B and additional experiment-specific information is provided at the appropriate places.

## 4    $\mathrm{A}d\mathrm{ANNS}$ – Adaptive ANNS

In this section, we present our proposed $\mathrm{A}d\mathrm{ANNS}$ 🏃 framework that exploits the inherent flexibility of matryoshka representations to improve the accuracy-compute trade-off for semantic search components. Standard ANNS pipeline can be split into two key components: (a) search data structure that indexes and stores data points, (b) query-point computation method that outputs (approximate) distance between a given query and data point. For example, standard IVFOPQ [24] method uses an IVF structure to index points on full-precision vectors and then relies on OPQ for more efficient distance computation between the query and the data points during the linear scan.

Below, we show that AdANNS can be applied to both the above-mentioned ANNS components and provides significant gains on the computation-accuracy tradeoff curve. In particular, we present AdANNS-IVF which is AdANNS version of the standard IVF index structure [48], and the closely related ScaNN structure [16]. We also present AdANNS-OPQ which introduces representation adaptivity in the OPQ, an industry-default quantization. Then, in Section 4.3 we further demonstrate the combination of the two techniques to get AdANNS-IVFOPQ – an AdANNS version of IVFOPQ [24] – and AdANNS-DiskANN, a similar variant of DiskANN [22]. Overall, our experiments show that AdANNS-IVF is significantly more accuracy-compute optimal compared to the IVF indices built on RRs and AdANNS-OPQ is as accurate as the OPQ on RRs while being significantly cheaper.

## 4.1 AdANNS-IVF

Recall from Section 1 that IVF has a clustering and a linear scan phase, where both phase use same dimensional rigid representation. Now, AdANNS-IVF allows the clustering phase to use the first $d_c$ dimensions of the given matryoshka representation (MR). Similarly, the linear scan within each cluster uses $d_s$ dimensions, where again $d_s$ represents top $d_s$ coordinates from MR. Note that setting $d_c = d_s$ results in non-adaptive regular IVF. Intuitively, we would set $d_c \ll d_s$, so that instead of clustering with a high-dimensional representation, we can approximate it accurately with a low-dimensional embedding of size $d_c$ followed by a linear scan with a higher $d_s$-dimensional representation. Intuitively, this helps in the smooth search of design space for state-of-the-art accuracy-compute

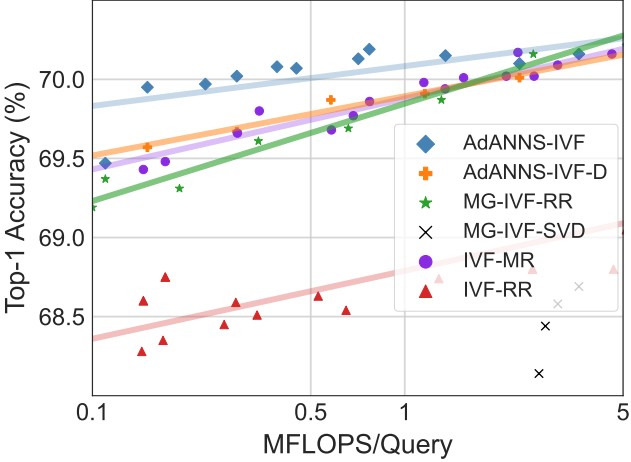

Figure 2: 1-NN accuracy on ImageNet retrieval shows that AdANNS-IVF achieves near-optimal accuracy-compute trade-off compared across various rigid and adaptive baselines. Both adaptive variants of MR and RR significantly outperform their rigid counterparts (IVF-XX) while post-hoc compression on RR using SVD for adaptivity falls short.

trade-off. Furthermore, this can provide a precise operating point on accuracy-compute tradeoff curve which is critical in several practical settings.

Our experiments on regular IVF with MRs and RRs (IVF-MR & IVF-RR) of varying dimensionalities and IVF configurations (# clusters, # probes) show that (Figure 2) matryoshka representations result in a significantly better accuracy-compute trade-off. We further studied and found that learned lower-dimensional representations offer better accuracy-compute trade-offs for IVF than higher-dimensional embeddings (see Appendix E for more results).

AdANNS utilizes $d$-dimensional matryoshka representation to get accurate $d_c$ and $d_s$ dimensional vectors at no extra compute cost. The resulting AdANNS-IVF provides a much better accuracy-compute trade-off (Figure 2) on ImageNet-1K retrieval compared to IVF-MR, IVF-RR, and MG-IVF-RR – multi-granular IVF with rigid representations (akin to AdANNS without MR) – a strong baseline that uses $d_c$ and $d_s$ dimensional RRs. Finally, we exhaustively search the design space of IVF by varying $d_c, d_s \in [8, 16, \dots, 2048]$ and the number of clusters $k \in [8, 16, \dots, 2048]$. Please see Appendix E for more details. For IVF experiments on the NQ dataset, please refer to Appendix G.

**Empirical results.** Figure 2 shows that AdANNS-IVF outperforms the baselines across all accuracy-compute settings for ImageNet-1K retrieval. AdANNS-IVF results in $10\times$ lower compute for the best accuracy of the extremely expensive MG-IVF-RR and non-adaptive IVF-MR. Specifically, as shown in Figure 1a, AdANNS-IVF is up to $1.5\%$ more accurate for the same compute and has up to $100\times$ lesser FLOPS/query ($90\times$ real-world speed-up!) than the status quo ANNS on rigid representations (IVF-RR). We filter out points for the sake of presentation and encourage the reader to check out Figure 8 in Appendix E for an expansive plot of all the configurations searched.

The advantage of AdANNS for construction of search structures is evident from the improvements in IVF (AdANNS-IVF) and can be easily extended to other ANNS structures like ScaNN [16] and

HNSW [38]. For example, HNSW consists of multiple layers with graphs of NSW graphs [37] of increasing complexity. AdANNS can be adopted to HNSW, where the construction of each level can be powered by appropriate dimensionalities for an optimal accuracy-compute trade-off. In general, AdANNS provides fine-grained control over compute overhead (storage, working memory, inference, and construction cost) during construction and inference while providing the best possible accuracy.

## 4.2 AdANNS-**OPQ**

Standard Product Quantization (PQ) essentially performs block-wise vector quantization via clustering. For example, suppose we need 32-byte PQ compressed vectors from the given 2048 dimensional representations. Then, we can chunk the representations in $m = 32$ equal blocks/sub-vectors of 64-d each, and each sub-vector space is clustered into $2^8 = 256$ partitions. That is, the representation of each point is essentially cluster-id for each block. Optimized PQ (OPQ) [13] further refines this idea, by first rotating the representations using a learned orthogonal matrix, and then applying PQ on top of the rotated representations. In ANNS, OPQ is used extensively to compress vectors and improves approximate distance computation primarily due to significantly lower memory overhead than storing full-precision data points IVF.

AdANNS-OPQ utilizes MR representations to apply OPQ on lower-dimensional representations. That is, for a given quantization budget, AdANNS allows using top $d_s \ll d$ dimensions from MR and then computing clusters with $d_s/m$-dimensional blocks where $m$ is the number of blocks. Depending on $d_s$ and $m$, we have further flexibility of trading-off dimensionality/capacity for increasing the number of clusters to meet the given quantization budget. AdANNS-OPQ tries multiple $d_s$, $m$, and number of clusters for a fixed quantization budget to obtain the best performing configuration.

We experimented with $8 - 128$ byte OPQ budgets for both ImageNet and Natural Questions retrieval with an exhaustive search on the quantized vectors. We compare AdANNS-OPQ which uses MRs of varying granularities to the baseline OPQ built on the highest dimensional RRs. We also evaluate OPQ vectors obtained projection using SVD [14] on top of the highest-dimensional RRs.

**Empirical results.** Figures 3 and 1b show that AdANNS-OPQ significantly outperforms – up to $4\%$ accuracy gain – the baselines (OPQ on RRs) across compute budgets on both ImageNet and NQ. In particular, AdANNS-OPQ tends to match the accuracy of a 64-byte (a typical choice in ANNS) OPQ baseline with only a 32-byte budget. This results in a $2\times$ reduction in both storage and compute FLOPS which translates to significant gains in real-world web-scale deployment (see Appendix D).

We only report the best AdANNS-OPQ for each budget typically obtained through a much lower-dimensional MR (128 & 192; much faster to build as well) than the highest-dimensional MR (2048 & 768) for ImageNet and NQ respectively (see Appendix G for more details). At the same time, we

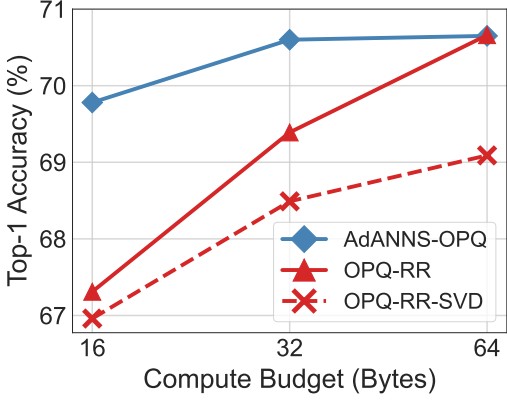

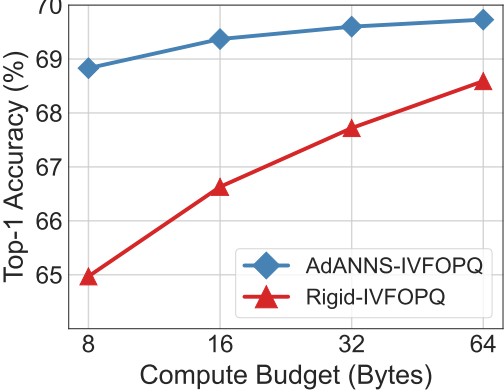

Figure 3: AdANNS-OPQ matches the accuracy of 64-byte OPQ on RR using only 32-bytes for ImageNet retrieval. AdANNS provides large gains at lower compute budgets and saturates to baseline performance for larger budgets.

Figure 4: Combining the gains of AdANNS for IVF and OPQ leads to better IVFOPQ composite indices. On ImageNet retrieval, AdANNS-IVFOPQ is $8\times$ cheaper for the same accuracy and provides 1 - $4\%$ gains over IVFOPQ on RRs.

note that building compressed OPQ vectors on projected RRs using SVD to the smaller dimensions (or using low-dimensional RRs, see Appendix D) as the optimal AdANNS-OPQ does not help in improving the accuracy. The significant gains we observe in AdANNS-OPQ are purely due to better information packing in MRs – we hypothesize that packing the most important information in the initial coordinates results in a better PQ quantization than RRs where the information is uniformly distributed across all the dimensions [31, 49]. See Appendix D for more details and experiments.

## 4.3 AdANNS for Composite Indices

We now extend AdANNS to composite indices [24] which put together two main ANNS building blocks – search structures and quantization – together to obtain efficient web-scale ANNS indices used in practice. A simple instantiation of a composite index would be the combination of IVF and OPQ – IVFOPQ – where the clustering in IVF happens with full-precision real vectors but the linear scan within each cluster is approximated using OPQ-compressed variants of the representation – since often the full-precision vectors of the database cannot fit in RAM. Contemporary ANNS indices like DiskANN [22] make this a default choice where they build the search graph with a full-precision vector and approximate the distance computations during search with an OPQ-compressed vector to obtain a very small shortlist of retrieved datapoints. In DiskANN, the shortlist of data points is then re-ranked to form the final list using their full-precision vectors fetched from the disk. AdANNS is naturally suited to this shortlist-rerank framework: we use a low-$d$ MR for forming index, where we could tune AdANNS parameters according to the accuracy-compute trade-off of the graph and OPQ vectors. We then use a high-$d$ MR for re-ranking.

**Empirical results.** Figure 4 shows that AdANNS-IVFOPQ is $1 - 4\%$ better than the baseline at all the PQ compute budgets. Furthermore, AdANNS-IVFOPQ has the same accuracy as the baselines at $8\times$ lower overhead. With DiskANN, AdANNS accelerates shortlist generation by using low-dimensional representations and recoups the accuracy by reranking with the highest-dimensional

Table 1: AdANNS-DiskANN using a $16$-$d$ MR + re-ranking with the $2048$-$d$ MR outperforms DiskANN built on $2048$-$d$ RR at *half* the compute cost on ImageNet retrieval.

|  | RR-2048 | AdANNS |
|---|---|---|
| PQ Budget (Bytes) | 32 | **16** |
| Top-1 Accuracy (%) | 70.37 | **70.56** |
| mAP@10 (%) | 62.46 | **64.70** |
| Precision@40 (%) | 65.65 | **68.25** |

MR at negligible cost. Table 1 shows that AdANNS-DiskANN is more accurate than the baseline for both 1-NN and ranking performance at only $half$ the cost. Using low-dimensional representations further speeds up inference in AdANNS-DiskANN (see Appendix F).

These results show the generality of AdANNS and its broad applicability across a variety of ANNS indices built on top of the base building blocks. Currently, AdANNS piggybacks on typical ANNS pipelines for their inherent accounting of the real-world system constraints [16, 22, 25]. However, we believe that AdANNS's flexibility and significantly better accuracy-compute trade-off can be further informed by real-world deployment constraints. We leave this high-potential line of work that requires extensive study to future research.

## 5 Further Analysis and Discussion

### 5.1 Compute-aware Elastic Search During Inference

AdANNS search structures cater to many specific large-scale use scenarios that need to satisfy precise resource constraints during construction as well as inference. However, in many cases, construction and storage of the indices are not the bottlenecks or the user is unable to search the design space. In these settings, AdANNS-D enables adaptive inference through accurate yet cheaper distance computation using the low-dimensional prefix of matryoshka representation. Akin to composite indices (Section 4.3) that use PQ vectors for cheaper distance computation, we can use the low-dimensional MR for faster distance computation on ANNS structure built *non-adaptively* with a high-dimensional MR without any modifications to the existing index.

**Empirical results.** Figure 2 shows that for a given compute budget using IVF on ImageNet-1K retrieval, AdANNS-IVF is better than AdANNS-IVF-D due to the explicit control during the building

of the ANNS structure which is expected. However, the interesting observation is that AdANNS-D *matches or outperforms* the IVF indices built with MRs of varying capacities for ImageNet retrieval.

However, these methods are applicable in specific scenarios of deployment. Obtaining optimal AdANNS search structure (highly accurate) or even the best IVF-MR index relies on a relatively expensive design search but delivers indices that fit the storage, memory, compute, and accuracy constraints all at once. On the other hand AdANNS-D does not require a precisely built ANNS index but can enable compute-aware search during inference. AdANNS-D is a great choice for setups that can afford only one single database/index but need to cater to varying deployment constraints, e.g., one task requires 70% accuracy while another task has a compute budget of 1 MFLOPS/query.

## 5.2 Why MRs over RRs?

Quite a few of the gains from AdANNS are owing to the quality and capabilities of matryoshka representations. So, we conducted extensive analysis to understand why matryoshka representations seem to be more aligned for semantic search than the status-quo rigid representations.

**Difficulty of NN search.** Relative contrast ($C_r$) [18] is inversely proportional to the difficulty of nearest neighbor search on a given database. On ImageNet-1K, Figure 14 shows that MRs have better $C_r$ than RRs across dimensionalities, further supporting that matryoshka representations are more aligned (easier) for NN search than existing rigid representations for the same accuracy. More details and analysis about this experiment can be found in Appendix H.2.

**Clustering distributions.** We also investigate the potential deviation in clustering distributions for MRs across dimensionalities compared to RRs. Unlike the RRs where the information is uniformly diffused across dimensions [49], MRs have hierarchical information packing. Figure 11 in Appendix E.3 shows that matryoshka representations result in clusters similar (measured by total variation distance [33]) to that of rigid representations and do not result in any unusual artifacts.

**Robustness.** Figure 9 in Appendix E shows that MRs continue to be better than RRs even for out-of-distribution (OOD) image queries (ImageNetV2 [44]) using ANNS. It also shows that the highest data dimensionality need not always be the most robust which is further supported by the higher recall using lower dimensions. Further details about this experiment can be found in Appendix E.1.

**Generality across encoders.** IVF-MR consistently has higher accuracy than IVF-RR across dimensionalities despite having similar accuracies with exact NN search (for ResNet50 on ImageNet and BERT-Base on NQ). We find that our observations on better alignment of MRs for NN search hold across neural network architectures, ResNet18/34/101 [19] and ConvNeXt-Tiny [35]. Appendix H.3 delves deep into the experimentation done using various neural architectures on ImageNet-1K.

**Recall score analysis.** Analysis of recall score (see Appendix C) in Appendix H.1 shows that for a similar top-1 accuracy, lower-dimensional representations have better 1-Recall@1 across search complexities for IVF and HNSW on ImageNet-1K. Across the board, MRs have higher recall scores and top-1 accuracy pointing to easier "searchability" and thus suitability of matryoshka representations for ANNS. Larger-scale experiments and further analysis can be found in Appendix H.

Through these analyses, we argue that matryoshka representations are better suited for semantic search than rigid representations, thus making them an ideal choice for AdANNS.

## 5.3 Search for AdANNS Hyperparameters

Choosing the optimal hyperparameters for AdANNS, such as $d_c$, $d_s$, $m$, # clusters, # probes, is an interesting and open problem that requires more rigorous examination. As the ANNS index is formed *once* and used for potentially billions of queries with massive implications for cost, latency and queries-per-second, a hyperparameter search for the best index is generally an acceptable industry practice [22, 38]. The Faiss library [24] provides guidelines[2] to choose the appropriate index for a specific problem, including memory constraints, database size, and the need for exact results. There have been efforts at automating the search for optimal indexing parameters, such as Autofaiss[3], which maximizes recall given compute constraints.

---

[2] https://github.com/facebookresearch/faiss/wiki/Guidelines-to-choose-an-index
[3] https://github.com/criteo/autofaiss

In case of AdANNS, we suggest starting at the best configurations of MRs followed by a local design space search to lead to near-optimal AdANNS configurations (e.g. use IVF-MR to bootstrap AdANNS-IVF). We also share some observations during the course of our experiments:

1. AdANNS-IVF: Top-1 accuracy generally improves (with diminishing returns after a point) with increasing dimensionality of clustering ($d_c$) and search ($d_s$), as we show on ImageNet variants and with multiple encoders in the Appendix (Figures 9 and 15). Clustering with low-$d$ MRs matches the performance of high-$d$ MRs as they likely contain similar amounts of useful information, making the increased compute cost not worth the marginal gains. Increasing # probes naturally boosts performance (Appendix, Figure 10a). Lastly, it is generally accepted that a good starting point for the # clusters $k$ is $\sqrt{N_D/2}$, where $N_D$ is the number of indexable items [39]. $k = \sqrt{N_D}$ is the optimal choice of $k$ from a FLOPS computation perspective as can be seen in Appendix B.1.

2. AdANNS-OPQ: we observe that for a fixed compute budget in bytes ($m$), the top-1 accuracy reaches a peak at $d < d_{max}$ (Appendix, Table 4). We hypothesize that the better performance of AdANNS-OPQ at $d < d_{max}$ is due to the curse of dimensionality, i.e. it is easier to learn PQ codebooks on smaller embeddings with similar amounts of information. We find that using an MR with $d = 4 \times m$ is a good starting point on ImageNet and NQ. We also suggest using an 8-bit (256-length) codebook for OPQ as the default for each of the sub-block quantizer.

3. AdANNS-DiskANN: Our observations with DiskANN are consistent with other indexing structures, i.e. the optimal graph construction dimensionality $d < d_{max}$ (Appendix, Figure 12). A careful study of DiskANN on different datasets is required for more general guidelines to choose graph construction and OPQ dimensionality $d$.

### 5.4 Limitations

AdANNS's core focus is to improve the design of the existing ANNS pipelines. To use AdANNS on a corpus, we need to back-fill [43] the MRs of the data – a significant yet a one-time overhead. We also notice that high-dimensional MRs start to degrade in performance when optimizing also for an extremely low-dimensional granularity (e.g., $< 24$-d for NQ) – otherwise is it quite easy to have comparable accuracies with both RRs and MRs. Lastly, the existing dense representations can only in theory be converted to MRs with an auto-encoder-style non-linear transformation. We believe most of these limitations form excellent future work to improve AdANNS further.

## 6 Conclusions

We proposed a novel framework, AdANNS 🏃, that leverages adaptive representations for different phases of ANNS pipelines to improve the accuracy-compute tradeoff. AdANNS utilizes the inherent flexibility of matryoshka representations [31] to design better ANNS building blocks than the standard ones which use the rigid representation in each phase. AdANNS achieves SOTA accuracy-compute trade-off for the two main ANNS building blocks: search data structures (AdANNS-IVF) and quantization (AdANNS-OPQ). The combination of AdANNS-based building blocks leads to the construction of better real-world composite ANNS indices – with as much as $8\times$ reduction in cost at the same accuracy as strong baselines – while also enabling compute-aware elastic search. Finally, we note that combining AdANNS with elastic encoders [11] enables truly adaptive large-scale retrieval.

### Acknowledgments

We are grateful to Kaifeng Chen, Venkata Sailesh Sanampudi, Sanjiv Kumar, Harsha Vardhan Simhadri, Gantavya Bhatt, Matthijs Douze and Matthew Wallingford for helpful discussions and feedback. Aditya Kusupati also thanks Tom Duerig and Rahul Sukthankar for their support. Part of the paper's large-scale experimentation is supported through a research GCP credit award from Google Cloud and Google Research. Sham Kakade acknowledges funding from the ONR award N00014-22-1-2377 and NSF award CCF-2212841. Ali Farhadi acknowledges funding from the NSF awards IIS 1652052, IIS 17303166, DARPA N66001-19-2-4031, DARPA W911NF-15-1-0543, and gifts from Allen Institute for Artificial Intelligence and Google.

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

# A AdANNS Framework

---

**Algorithm 1** AdANNS-IVF Psuedocode

---

```
# Index database to construct clusters and build inverted file system

def adannsConstruction(database, d_cluster, num_clusters):
    # Slice database with cluster construction dim (d_cluster)
    xb = database[:d_cluster]
    cluster_centroids = constructClusters(xb, num_clusters)

    return cluster_centroids

def adannsInference(queries, centroids, d_shortlist, d_search, num_probes,
    k):
    # Slice queries and centroids with cluster shortlist dim (d_shortlist)
    xq = queries[:d_shortlist]
    xc = centroids[:d_shortlist]

    for q in queries:
        # compute distance of query from each cluster centroid
        candidate_distances = computeDistances(q, xc)
        # sort cluster candidates by distance and choose small number to
            probe
        cluster_candidates = sortAscending(candidate_distances)[:num_probes]
        database_candidates = getClusterMembers(cluster_candidates)
        # Linear Scan all shortlisted clusters with search dim (d_search)
        k_nearest_neighbors[q] = linearScan(q, database_candidates, d_search,
            k)

    return k_nearest_neighbors
```

---

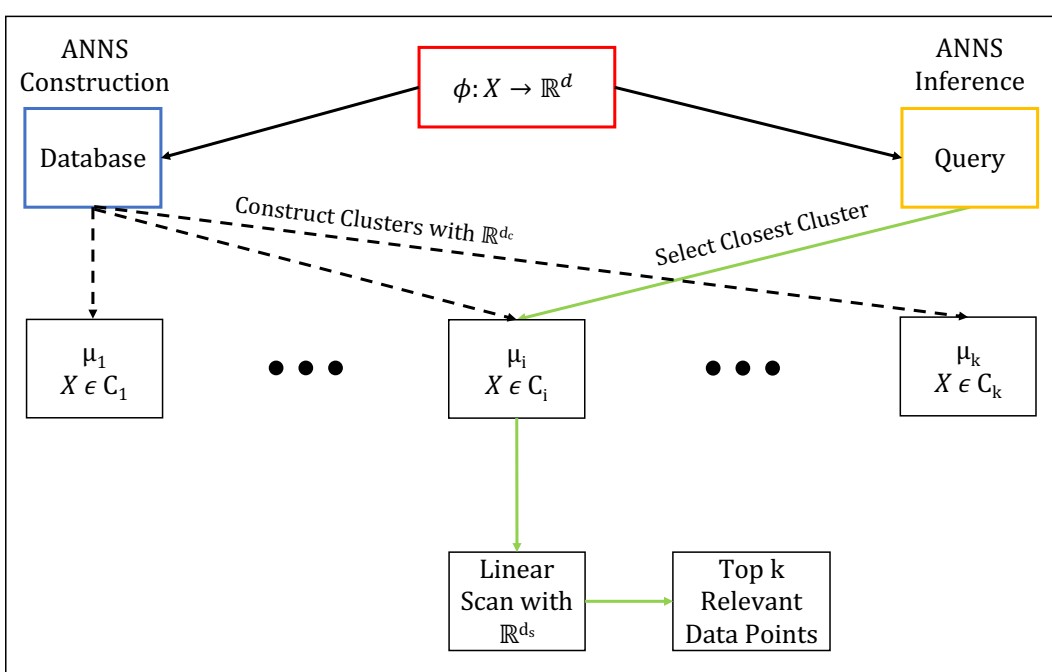

Figure 5: The schematic of inverted file index (IVF) outlaying the construction and inference phases. Adaptive representations can be utilized effectively in the decoupled components of clustering and searching for a better accuracy-compute trade-off (AdANNS-IVF).

Table 2: Mathematical formulae of the retrieval phase across various methods built on IVF. See Section 3 for notations.

| Method | Retrieval Formula during Inference |
|---|---|
| IVF-RR | $\arg\min_{j \in C_{h(q)}} \|\phi^{\mathrm{RR}(d)}(q) - \phi^{\mathrm{RR}(d)}(x_j)\|$, s.t. $h(q) = \arg\min_h \|\phi^{\mathrm{RR}(d)}(q) - \mu_h^{\mathrm{RR}(d)}\|$ |
| IVF-MR | $\arg\min_{j \in C_{h(q)}} \|\phi^{\mathrm{MR}(d)}(q) - \phi^{\mathrm{MR}(d)}(x_j)\|$, s.t. $h(q) = \arg\min_h \|\phi^{\mathrm{MR}(d)}(q) - \mu_h^{\mathrm{MR}(d)}\|$ |
| AdANNS-IVF | $\arg\min_{j \in C_{h(q)}} \|\phi^{\mathrm{MR}(d_s)}(q) - \phi^{\mathrm{MR}(d_s)}(x_j)\|$, s.t. $h(q) = \arg\min_h \|\phi^{\mathrm{MR}(d_c)}(q) - \mu_h^{\mathrm{MR}(d_c)}\|$ |
| MG-IVF-RR | $\arg\min_{j \in C_{h(q)}} \|\phi^{\mathrm{RR}(d_s)}(q) - \phi^{\mathrm{RR}(d_s)}(x_j)\|$, s.t. $h(q) = \arg\min_h \|\phi^{\mathrm{RR}(d_c)}(q) - \mu_h^{\mathrm{RR}(d_c)}\|$ |
| AdANNS-IVF-D | $\arg\min_{j \in C_{h(q)}} \|\phi^{\mathrm{MR}(d)}(q)[1:\hat{d}] - \phi^{\mathrm{MR}(d)}(x_j)[1:\hat{d}]\|$, s.t. $h(q) = \arg\min_h \|\phi^{\mathrm{MR}(d)}(q)[1:\hat{d}] - \mu_h^{\mathrm{MR}(d)}[1:\hat{d}]\|$ |
| IVFOPQ | $\arg\min_{j \in C_{h(q)}} \|\phi^{\mathrm{PQ}(m,b)}(q) - \phi^{\mathrm{PQ}(m,b)}(x_j)\|$, s.t. $h(q) = \arg\min_h \|\phi(q) - \mu_h\|$ |

# B  Training and Compute Costs

A bulk of our ANNS experimentation was written with Faiss [24], a library for efficient similarity search and clustering. AdANNS was implemented from scratch (Algorithm 1) due to difficulty in decoupling clustering and linear scan with Faiss, with code available at https://github.com/RAIVNLab/AdANNS. We also provide a version of AdANNS with Faiss optimizations with the restriction that $D_c \geq D_s$ as a limitation of the current implementation, which can be further optimized. All ANNS experiments (AdANNS-IVF, MG-IVF-RR, IVF-MR, IVF-RR, HNSW, HNSWOPQ, IVFOPQ) were run on an Intel Xeon 2.20GHz CPU with 12 cores. Exact Search (Flat L2, PQ, OPQ) and DiskANN experiments were run with CUDA 11.0 on a A100-SXM4 NVIDIA GPU with 40G RAM. The wall-clock inference times quoted in Figure 1a and Table 3 are reported on CPU with Faiss optimizations, and are averaged over three inference runs for ImageNet-1K retrieval.

Table 3: Comparison of AdANNS-IVF and Rigid-IVF wall-clock inference times for ImageNet-1K retrieval. AdANNS-IVF has up to $\sim 1.5\%$ gain over Rigid-IVF for a fixed search latency per query.

| AdANNS-IVF | | Rigid-IVF | |
|---|---|---|---|
| Top-1 | Search Latency/Query (ms) | Top-1 | Search Latency/Query (ms) |
| 70.02 | 0.03 | 68.51 | 0.02 |
| 70.08 | 0.06 | 68.54 | 0.05 |
| 70.19 | 0.06 | 68.74 | 0.08 |
| 70.36 | 0.88 | 69.20 | 0.86 |
| 70.60 | 5.57 | 70.13 | 5.67 |

**DPR [27] on NQ [32].** We follow the setup on the DPR repo[4]: the Wikipedia corpus has 21 million passages and Natural Questions dataset for open-domain QA settings. The training set contains 79,168 question and answer pairs, the dev set has 8,757 pairs and the test set has 3,610 pairs.

## B.1  Inference Compute Cost

We evaluate inference compute costs for IVF in MegaFLOPS per query (MFLOPS/query) as shown in Figures 2, 10a, and 8 as follows:

$$C = d_s k + \frac{n_p d_s N_D}{k}$$

where $d_c$ is the **cluster** construction embedding dimensionality, $d_s$ is the embedding dim used for linear **scan** within each **probed** cluster, which is controlled by # of search probes $n_p$. Finally, $k$ is the number of clusters $|C_i|$ indexed over database of size $N_D$. The default setting in this work, unless otherwise stated, is $n_p = 1$, $k = 1024$, $N_D = 1281167$ (ImageNet-1K trainset). Vanilla IVF supports only $d_c = d_s$, while AdANNS-IVF provides flexibility via decoupling clustering and search (Section 4). AdANNS-IVF-D is a special case of AdANNS-IVF with the flexibility restricted to inference, i.e., $d_c$ is a fixed high-dimensional MR.

---

[4] https://github.com/facebookresearch/DPR

# C   Evaluation Metrics

In this work, we primarily use top-1 accuracy (i.e. 1-Nearest Neighbor), recall@k, corrected mean average precision (mAP@k) [30] and k-Recall@N (recall score), which are defined over all queries $Q$ over indexed database of size $N_D$ as:

$$\text{top-1} = \frac{\sum_Q \text{correct\_pred@1}}{|Q|}$$

$$\text{Recall@k} = \frac{\sum_Q \text{correct\_pred@}k}{|Q|} * \frac{\text{num\_classes}}{|N_D|}$$

where correct_pred@$k$ is the number of k-NN with correctly predicted labels for a given query. As noted in Section 3, k-Recall@N is the overlap between $k$ exact search nearest neighbors (considered as ground truth) and the top N retrieved documents. As Faiss [24] supports a maximum of 2048-NN while searching the indexed database, we report 40-Recall@2048 in Figure 13. Also note that for ImageNet-1K, which constitutes a bulk of the experimentation in this work, $|Q| = 50000$, $|N_D| = 1281167$ and num_classes = 1000. For ImageNetv2 [44], $|Q| = 10000$ and num_classes = 1000, and for ImageNet-4K [31], $|Q| = 210100$, $|N_D| = 4202000$ and num_classes = 4202. For NQ [32], $|Q| = 3610$ and $|N_D| = 21015324$. As NQ consists of question-answer pairs (instance-level), num_classes = 3610 for the test set.

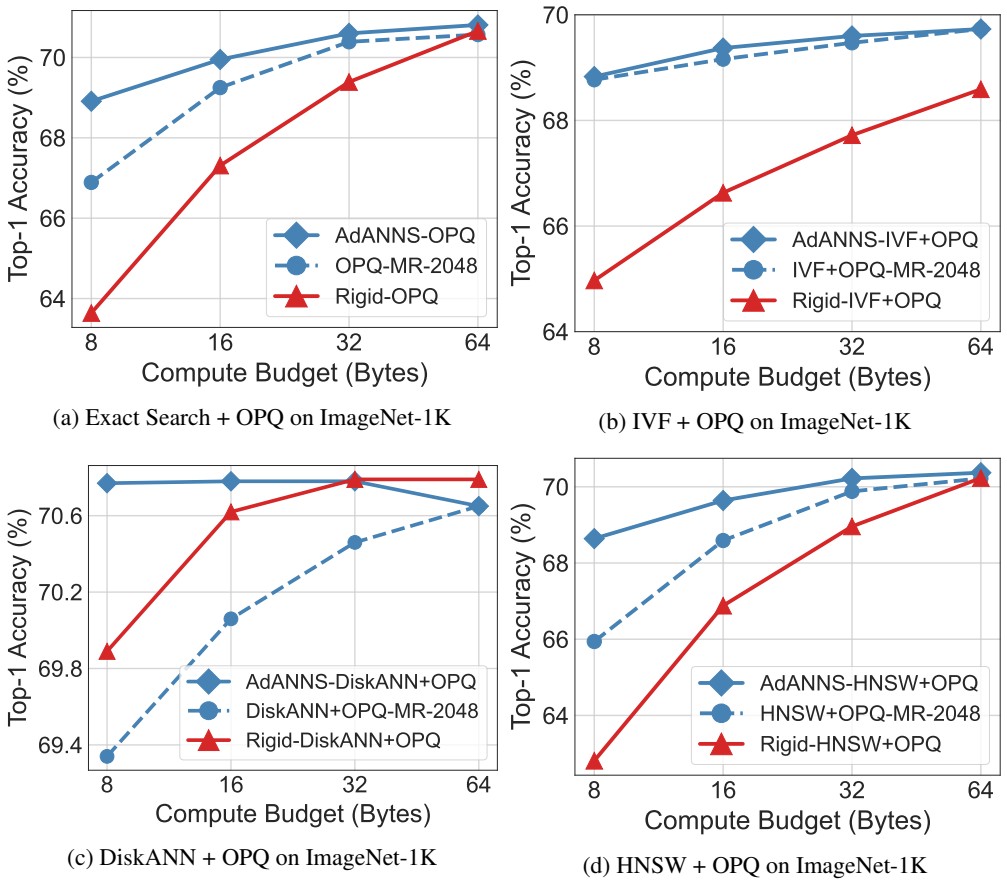

Figure 6: Top-1 Accuracy of AdANNS composite indices with OPQ distance computation compared to MR and Rigid baselines models on ImageNet-1K and Natural Questions.

## D   AdANNS-OPQ

In this section, we take a deeper dive into the quantization characteristics of MR. In this work, we restrict our focus to optimized product quantization (OPQ) [13], which adds a learned space rotation and dimensionality permutation to PQ's sub-vector quantization to learn more optimal PQ codes. We compare OPQ to vanilla PQ on ImageNet in Table 4, and observe large gains at larger embedding dimensionalities, which agrees with the findings of Jayaram Subramanya et al. [22].

We perform a study of composite OPQ $m \times b$ indices on ImageNet-1K across compression compute budgets $m$ (where $b = 8$, i.e. 1 Byte), i.e. Exact Search with OPQ, IVF+OPQ, HNSW+OPQ, and DiskANN+OPQ, as seen in Figure 6. It is evident from these results:

1. Learning OPQ codebooks with AdANNS (Figure 6a) provides a 1-5% gain in top-1 accuracy over rigid representations at low compute budgets ($\leq$ 32 Bytes). AdANNS-OPQ saturates to Rigid-OPQ performance at low compression ($\geq$ 64 Bytes).

2. For IVF, learning clusters with MRs instead of RRs (Figure 6b) provides substantial gains (1-4%). In contrast to Exact-OPQ, using AdANNS for learning OPQ codebooks does not provide substantial top-1 accuracy gains over MR with $d = 2048$ (highest), though it is still slightly better or equal to MR-2048 at all compute budgets. This further supports that IVF performance generally scales with embedding dimensionality, which is consistent with our findings on ImageNet across robustness variants and encoders (See Figures 9 and 15 respectively).

3. Note that in contrast to Exact, IVF, and HNSW coarse quantizers, DiskANN *inherently re-ranks* the retrieved shortlist with high-precision embeddings ($d = 2048$), which is reflected in its high top-1 accuracy. We find that AdANNS with 8-byte OPQ (Figure 6c) matches the top-1 accuracy of rigid representations using 32-byte OPQ, for a $4\times$ cost reduction for the same accuracy. Also note that using AdANNS provides large gains over using MR-2048 at high compression (1.5%), highlighting the necessity of AdANNS's flexibility for high-precision retrieval at low compute budgets.

4. Our findings on the HNSW-OPQ composite index (Figure 6d) are consistent with all other indices, i.e. HNSW graphs constructed with AdANNS OPQ codebooks provide significant gains over RR and MR, especially at high compression ($\leq$ 32 Bytes).

**OPQ on NQ dataset**

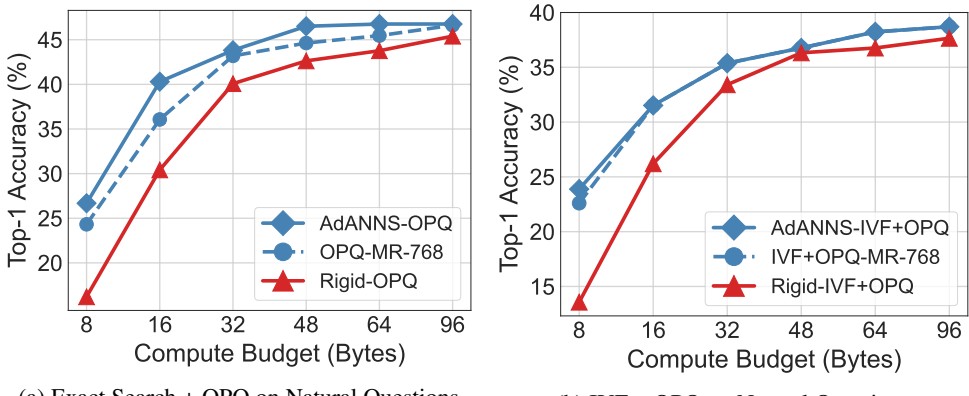

(a) Exact Search + OPQ on Natural Questions    (b) IVF + OPQ on Natural Questions

Figure 7: Top-1 Accuracy of AdANNS composite indices with OPQ distance computation compared to MR and Rigid baselines models on Natural Questions.

Our observations on ImageNet with ResNet-50 MR across search structures also extend to the Natural Questions dataset with Dense Passage Retriever (DPR with BERT-Base MR embeddings). We note that AdANNS provides gains over RR-768 embeddings for both Exact Search and IVF with OPQ (Figure 7a and 7b). We find that similar to ImageNet (Figure 15) IVF performance on Natural Questions generally scales with dimensionality. AdANNS thus reduces to MR-768 performance for $M \geq 16$. See Appendix G for a more in-depth discussion of AdANNS with DPR on Natural Questions.

Table 4: Comparison of PQ-MR with OPQ-MR for exact search on ImageNet-1K across embedding dimensionality $d \in \{8, 16, ..., 2048\}$ quantized to $m \in \{8, 16, 32, 64\}$ bytes. OPQ shows large gains over vanilla PQ at larger embedding dimensionalities $d \geq 128$. Entries with the highest top-1 accuracy for a given $(d, m)$ tuple are bolded.

| Config | | PQ | | | OPQ | | |
|---|---|---|---|---|---|---|---|
| d | m | Top-1 | mAP@10 | P@100 | Top-1 | mAP@10 | P@100 |
| 8 | 8 | 62.18 | 56.71 | 61.23 | **62.22** | 56.70 | 61.23 |
| 16 | 8 | **67.91** | 62.85 | 67.21 | 67.88 | 62.96 | 67.21 |
| | 16 | 67.85 | 62.95 | 67.21 | **67.96** | 62.94 | 67.21 |
| 32 | 8 | 68.80 | 63.62 | 67.86 | **68.91** | 63.63 | 67.86 |
| | 16 | **69.57** | 64.22 | 68.12 | 69.47 | 64.20 | 68.12 |
| | 32 | 69.44 | 64.20 | 68.12 | **69.47** | 64.23 | 68.12 |
| 64 | 8 | **68.39** | 63.40 | 67.47 | 68.38 | 63.42 | 67.60 |
| | 16 | 69.77 | 64.43 | 68.25 | **69.95** | 64.55 | 68.38 |
| | 32 | **70.13** | 64.67 | 68.38 | 70.05 | 64.65 | 68.38 |
| | 64 | 70.12 | 64.69 | 68.42 | **70.18** | 64.70 | 68.38 |
| 128 | 8 | 67.27 | 61.99 | 65.78 | **68.40** | 63.11 | 67.34 |
| | 16 | 69.51 | 64.32 | 68.12 | **69.78** | 64.56 | 68.38 |
| | 32 | 70.27 | 64.72 | 68.51 | **70.60** | 64.97 | 68.51 |
| | 64 | 70.61 | 64.93 | 68.49 | **70.65** | 64.98 | 68.51 |
| 256 | 8 | 66.06 | 60.44 | 64.09 | **67.90** | 62.69 | 66.95 |
| | 16 | 68.56 | 63.33 | 66.95 | **69.92** | 64.71 | 68.51 |
| | 32 | 70.08 | 64.83 | 68.38 | **70.59** | 65.15 | 68.64 |
| | 64 | 70.48 | 64.98 | 68.55 | **70.69** | 65.09 | 68.64 |
| 512 | 8 | 65.09 | 59.03 | 62.53 | **67.51** | 62.12 | 66.56 |
| | 16 | 67.68 | 62.11 | 65.39 | **69.67** | 64.53 | 68.38 |
| | 32 | 69.51 | 64.01 | 67.34 | **70.44** | 65.11 | 68.64 |
| | 64 | 70.53 | 65.02 | 68.52 | **70.72** | 65.17 | 68.64 |
| 1024 | 8 | 64.58 | 58.26 | 61.75 | **67.26** | 62.07 | 66.56 |
| | 16 | 66.84 | 61.07 | 64.09 | **69.34** | 64.23 | 68.12 |
| | 32 | 68.71 | 62.92 | 66.04 | **70.43** | 65.03 | 68.64 |
| | 64 | 69.88 | 64.35 | 67.68 | **70.81** | 65.19 | 68.64 |
| 2048 | 8 | 62.19 | 56.11 | 59.80 | **66.89** | 61.69 | 66.30 |
| | 16 | 65.99 | 60.27 | 63.18 | **69.25** | 64.09 | 67.99 |
| | 32 | 67.99 | 62.04 | 64.74 | **70.39** | 64.97 | 68.51 |
| | 64 | 69.20 | 63.46 | 66.40 | **70.57** | 65.15 | 68.51 |

# E   AdANNS-IVF

Inverted file index (IVF) [48] is a simple yet powerful ANNS data structure used in web-scale search systems [16]. IVF construction involves clustering (coarse quantization often through k-means) [36] on $d$-dimensional representation that results in an inverted file list [53] of all the data points in each cluster. During search, the $d$-dimensional query representation is first assigned to the closest clusters (# probes, typically set to 1) and then an exhaustive linear scan happens within each cluster to obtain the nearest neighbors. As seen in Figure 9, IVF top-1 accuracy scales logarithmically with increasing representation dimensionality $d$ on ImageNet-1K/V2/4K. The learned low-$d$ representations thus provide better accuracy-compute trade-offs compared to high-$d$ representations, thus furthering the case for usage of AdANNS with IVF.

Our proposed adaptive variant of IVF, AdANNS-IVF, decouples the clustering, with $d_c$ dimensions, and the linear scan within each cluster, with $d_s$ dimensions – setting $d_c = d_s$ results in non-adaptive vanilla IVF. This helps in the smooth search of design space for the optimal accuracy-compute trade-off. A naive instantiation yet strong baseline would be to use explicitly trained $d_c$ and $d_s$ dimensional rigid representations (called MG-IVF-RR, for multi-granular IVF with

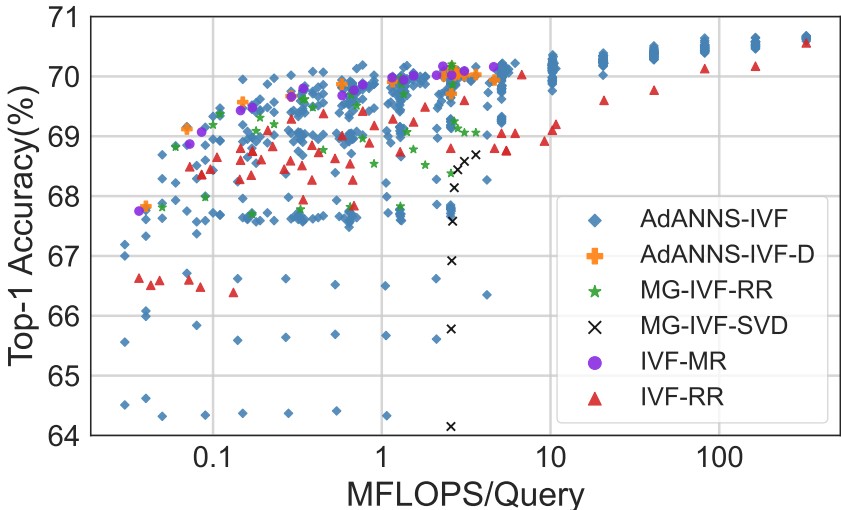

Figure 8: Top-1 accuracy vs. compute cost per query of AdANNS-IVF compared to IVF-MR, IVF-RR and MG-IVF-RR baselines on ImageNet-1K.

rigid representations). We also examine the setting of adaptively choosing low-dimensional MR to linear scan the shortlisted clusters built with high-dimensional MR, i.e. AdANNS-IVF-D, as seen in Table 5. As seen in Figure 8, AdANNS-IVF provides pareto-optimal accuracy-compute tradeoff across inference compute. This figure is a more exhaustive indication of AdANNS-IVF behavior compared to baselines than Figures 1a and 2. AdANNS-IVF is evaluated for all possible tuples of $d_c, d_s, k = |C| \in \{8, 16, \dots, 2048\}$. AdANNS-IVF-D is evaluated for a pre-built IVF index with $d_c = 2048$ and $d_s \in \{8, \dots, 2048\}$. MG-IVF-RR configurations are evaluated for $d_c \in \{8, \dots, d_s\}, d_s \in \{32, \dots, 2048\}$ and $k = 1024$ clusters. A study over additional $k$ values is omitted due to high compute cost. Finally, IVF-MR and IVF-RR configurations are evaluated for $d_c = d_s \in \{8, 16, \dots, 2048\}$ and $k \in \{256, \dots, 8192\}$. Note that for a fair comparison, we use $n_p = 1$ across all configurations. We discuss the inference compute for these settings in Appendix B.1.

### E.1 Robustness

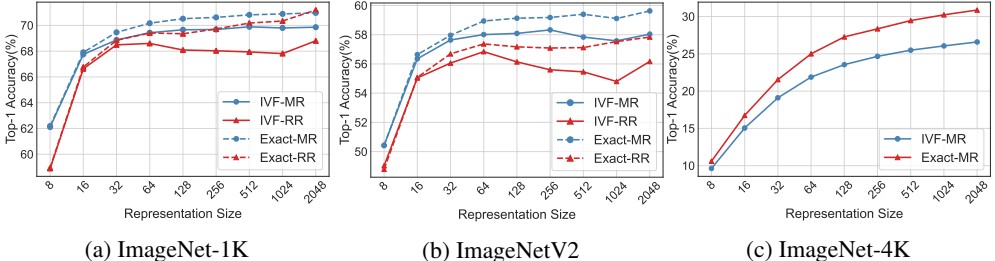

Figure 9: Top-1 Accuracy variation of IVF-MR of ImageNet 1K, ImageNetV2 and ImageNet-4K. RR baselines are omitted on ImageNet-4K due to high compute cost.

As shown in Figure 9, we examined the clustering capabilities of MRs on both in-distribution (ID) queries via ImageNet-1K and out-of-distribution (OOD) queries via ImageNetV2 [44], as well as on larger-scale ImageNet-4K [31]. For ID queries on ImageNet-1K (Figure 9a), IVF-MR is at least as accurate as Exact-RR for $d \le 256$ with a single search probe, demonstrating the quality of in-distribution low-d clustering with MR. On OOD queries (Figure 9b), we observe that IVF-MR is on average 2% more robust than IVF-RR across all cluster construction and linear scan dimensionalities $d$. It is also notable that clustering with MRs followed by linear scan with # probes $= 1$ is more robust than exact search with RR embeddings across all $d \le 2048$, indicating the adaptability of MRs to distribution shifts during inference. As seen in Table 5, on ImageNetV2 AdANNS-IVF-D is the best configuration for $d \le 16$, and is similarly accurate to IVF-MR at all other $d$. AdANNS-IVF-D with

Table 5: Top-1 Accuracy of AdANNS-IVF-D on out-of-distribution queries from ImageNetV2 compared to both IVF and Exact Search with MR and RR embeddings. Note that for AdANNS-IVF-D, the dimensionality used to build clusters $d_c = 2048$.

| d | AdANNS-IVF-D | IVF-MR | Exact-MR | IVF-RR | Exact-RR |
|---|---|---|---|---|---|
| 8 | **53.51** | 50.44 | 50.41 | 49.03 | 48.79 |
| 16 | **57.32** | 56.35 | 56.64 | 55.04 | 55.08 |
| 32 | 57.32 | 57.64 | **57.96** | 56.06 | 56.69 |
| 64 | 57.85 | 58.01 | **58.94** | 56.84 | 57.37 |
| 128 | 58.02 | 58.09 | **59.13** | 56.14 | 57.17 |
| 256 | 58.01 | 58.33 | **59.18** | 55.60 | 57.09 |
| 512 | 58.03 | 57.84 | **59.40** | 55.46 | 57.12 |
| 1024 | 57.66 | 57.58 | **59.11** | 54.80 | 57.53 |
| 2048 | 58.04 | 58.04 | **59.63** | 56.17 | 57.84 |

$d = 128$ is able to match its own accuracy with $d = 2048$, a $16\times$ compute gain during inference. This demonstrates the potential of AdANNS to adaptively search pre-indexed clustering structures.

On 4-million scale ImageNet-4K (Figure 9c), we observe similar accuracy trends of IVF-MR compared to Exact-MR as in ImageNet-1K (Figure 9a) and ImageNetV2 (Figure 9b). We omit baseline IVF-RR and Exact-RR experiments due to high compute cost at larger scale.

## E.2 IVF-MR Ablations

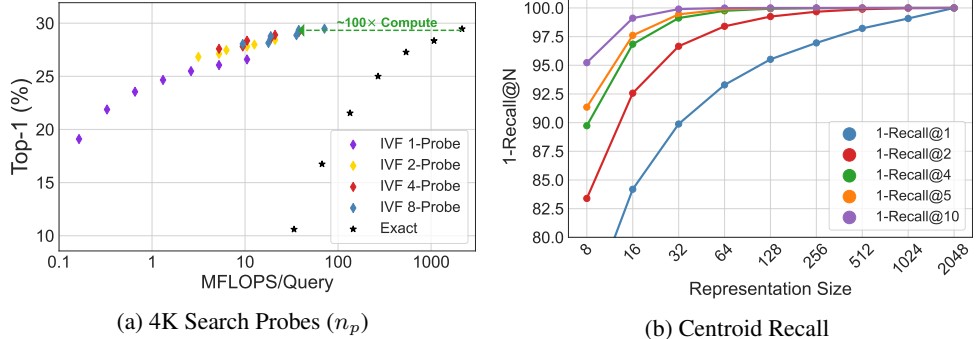

(a) 4K Search Probes $(n_p)$       (b) Centroid Recall

Figure 10: Ablations on IVF-MR Clustering: a) Analysis of accuracy-compute tradeoff with increasing IVF-MR search probes $n_p$ on ImageNet-4K compared to Exact-MR and b) k-Recall@N on ImageNet-1K cluster centroids across representation sizes $d$. Cluster centroids retrieved with highest embedding dim $d = 2048$ were considered ground-truth centroids.

As seen in Figure 10a, IVF-MR can match the accuracy of Exact Search on ImageNet-4K with $\sim 100\times$ less compute. We also explored the capability of MRs at retrieving cluster centroids with low-d compared to a ground truth of 2048-d with k-Recall@N, as seen in Figure 10b. MRs were able to saturate to near-perfect 1-Recall@N for $d \geq 32$ and $N \geq 4$, indicating the potential of AdANNS at matching exact search performance with less than 10 search probes $n_p$.

## E.3 Clustering Distribution

We examined the distribution of learnt clusters across embedding dimensionalities $d$ for both MR and RR models, as seen in Figure 11. We observe IVF-MR to have less variance than IVF-RR at $d \in \{8, 16\}$, and slightly higher variance for $d \geq 32$, while IVF-MR outperforms IVF-RR in top-1 across all $d$ (Figure 9a). This indicates that although MR learns clusters that are less uniformly distributed than RR at high $d$, the quality of learnt clustering is superior to RR across all $d$. Note that a uniform distribution is $N/k$ data points per cluster, i.e. $\sim 1250$ for ImageNet-1K with $k = 1024$. We quantitatively evaluate the proximity of the MR and RR clustering distributions with Total Variation

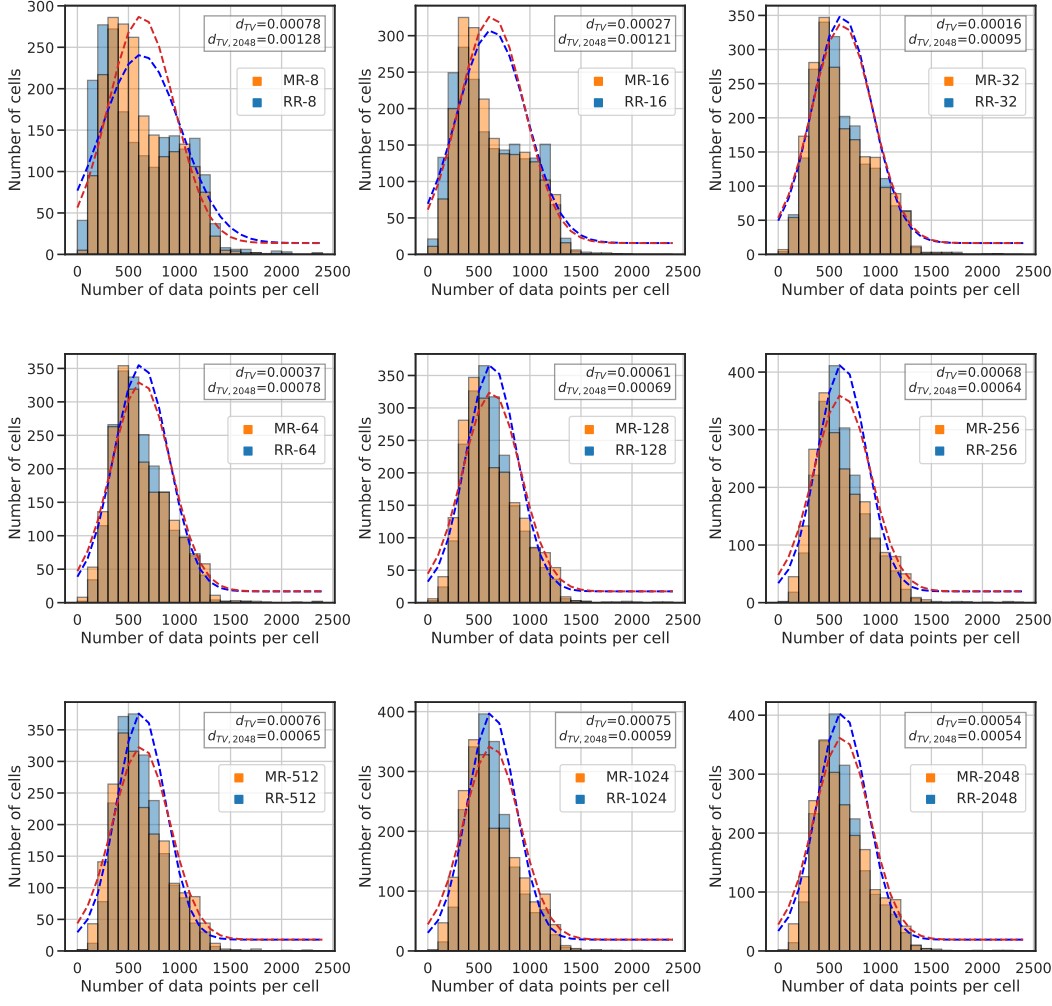

Figure 11: Clustering distributions for IVF-MR and IVF-RR across embedding dimensionality $d$ on ImageNet-1K. An IVF-MR and IVF-RR clustered with $d = 16$ embeddings is denoted by MR-16 and RR-16 respectively.

Distance [33], which is defined over two discrete probability distributions $p, q$ over $[n]$ as follows:

$$d_{TV}(p,q) = \frac{1}{2} \sum_{i \in [n]} |p_i - q_i|$$

We also compute $d_{TV,2048}(\text{MR-d}) = d_{TV}(\text{MR-d}, \text{RR-2048})$, which evaluates the total variation distance of a given low-d MR from high-d RR-2048. We observe a monotonically decreasing $d_{TV,2048}$ with increasing $d$, which demonstrates that MR clustering distributions get closer to RR-2048 as we increase the embedding dimensionality $d$. We observe in Figure 11 that $d_{TV}(\text{MR-d}, \text{RR-d}) \sim 7e - 4$ for $d \in \{8, 256, \dots, 2048\}$ and $\sim 3e - 4$ for $d \in \{16, 32, 64\}$. These findings agree with the top-1 improvement of MR over RR as shown in Figure 9a, where there are smaller improvements for $d \in \{16, 32, 64\}$ (smaller $d_{TV}$) and larger improvements for $d \in \{8, 256, \dots, 2048\}$ (larger $d_{TV}$). These results demonstrate a correlation between top-1 performance of IVF-MR and the quality of clusters learnt with MR.

## F AdANNS-DiskANN

DiskANN is a state-of-the-art graph-based ANNS index capable of serving queries from both RAM and SSD. DiskANN builds a greedy best-first graph with OPQ distance computation, with compressed vectors stored in memory. The index and full-precision vectors are stored on the SSD. During search,

Table 6: Wall clock search latency ($\mu s$) of AdANNS-DiskANN across graph construction dimensionality $d \in \{8, 16, \dots, 2048\}$ and compute budget in terms of OPQ budget $M \in \{8, 16, 32, 48, 64\}$. Search latency is fairly consistent across fixed embedding dimensionality $D$.

| $d$ | $M$=8 | $M$=16 | $M$=32 | $M$=48 | $M$=64 |
|---|---|---|---|---|---|
| 8 | 495 | - | - | - | - |
| 16 | 555 | 571 | - | - | - |
| 32 | 669 | 655 | 653 | - | - |
| 64 | 864 | 855 | 843 | 844 | 848 |
| 128 | 1182 | 1311 | 1156 | 1161 | 2011 |
| 256 | 1923 | 1779 | 1744 | 2849 | 1818 |
| 512 | 2802 | 3272 | 3423 | 2780 | 3171 |
| 1024 | 5127 | 5456 | 5724 | 4683 | 5087 |
| 2048 | 9907 | 9833 | 10205 | 10183 | 9329 |

when a query's neighbor shortlist is fetched from the SSD, its full-precision vector is also fetched in a single disk read. This enables efficient and fast distance computation with PQ on a large initial shortlist of candidate nearest neighbors in RAM followed by a high-precision re-ranking with full-precision vectors fetched from the SSD on a much smaller shortlist. The experiments carried out in this work primarily utilize a DiskANN graph index built in-memory[5] with OPQ distance computation.

As with IVF, DiskANN is also well suited to the flexibility provided by AdANNS as we demonstrate on both ImageNet and NQ that the optimal PQ codebook for a given compute budget is learnt with a smaller embedding dimensionality $d$ (see Figures 6c and 7a). We demonstrate the capability of AdANNS-DiskANN with a compute budget of $m \in \{32, 64\}$ in Table 1. We tabulate the search time latency of AdANNS-DiskANN in microseconds ($\mu s$) in Table 6, which grows linearly with graph construction dimensionality $d$. We also examine DiskANN-MR with SSD graph indices on ImageNet-1K across OPQ budgets for distance computation $m_{dc} \in \{32, 48, 64\}$, as seen in Figure 12. With SSD indices, we store PQ-compressed vectors on disk with $m_{disk} = m_{dc}$, which essentially disables DiskANN's implicit high-precision re-ranking. We ob-

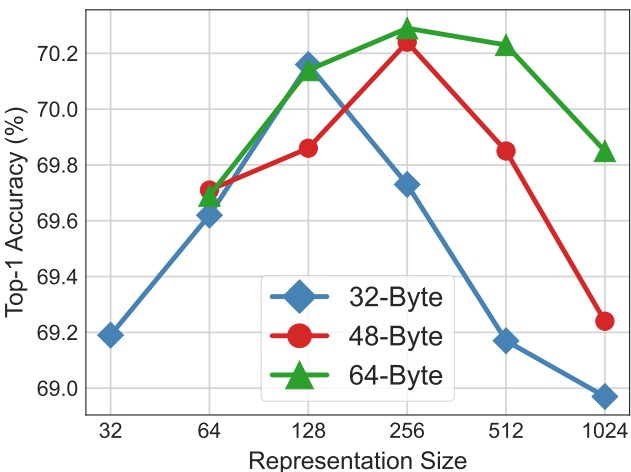

Figure 12: DiskANN-MR with SSD indices for ImageNet-1K retrieval, with compute budgets $m_{disk} = m_{dc} \in \{32, 48, 64\}$ across graph and OPQ codebook construction dimensionalities $d \in \{32, \dots, 1024\}$. Note that this does not use any re-ranking after obtaining OPQ based shortlist.

serve similar trends to other composite ANNS indices on ImageNet, where the *optimal* dim for fixed OPQ budget is not the highest dim ($d = 1024$ with fp32 embeddings is current highest dim supported by DiskANN which stores vectors in 4KB sectors on disk). This provides further motivation for AdANNS-DiskANN, which leverages MRs to provide flexible access to the optimal dim for quantization and thus enables similar Top-1 accuracy to Rigid DiskANN for up to $1/4$ the cost (Figure 6c).

# G   AdANNS on Natural Questions

In addition to image retrieval on ImageNet, we also experiment with dense passage retrieval (DPR) on Natural Questions. As shown in Figure 6, MR representations are $1 - 10\%$ more accurate than their

---
[5]https://github.com/microsoft/DiskANN

RR counterparts across PQ compute budgets with Exact Search + OPQ on NQ. We also demonstrate that IVF-MR is $1 - 2.5\%$ better than IVF-RR for Precision@$k$, $k \in \{1, 5, 20, 100, 200\}$. Note that on NQ, IVF loses $\sim 10\%$ accuracy compared to exact search, even with the RR-768 baseline. We hypothesize the weak performance of IVF owing to poor clusterability of the BERT-Base embeddings fine-tuned on the NQ dataset. A more thorough exploration of AdANNS-IVF on NQ is an immediate future work and is in progress.

## H    Ablations

### H.1    Recall Score Analysis

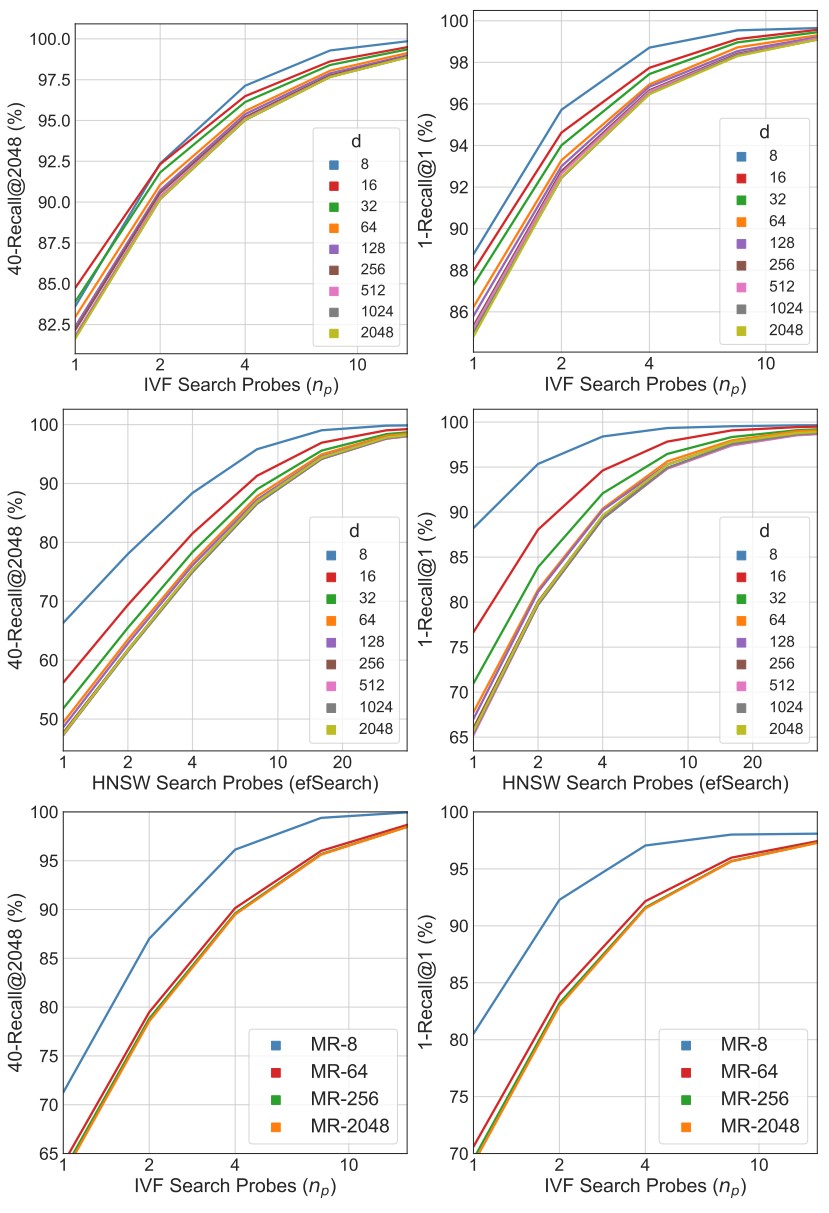

Figure 13: k-Recall@N of $d$-dimensional MR for IVF and HNSW with increasing search probes $n_p$ on ImageNet-1K and ImageNet-4K. On ImageNet-4K, we restrict our study to IVF-MR with $d \in \{8, 64, 256, 2048\}$. Other embedding dimensionalities, HNSW-MR and RR baselines are omitted due to high compute cost. We observe that trends from ImageNet-1K with increasing $d$ and $n_p$ extend to ImageNet-4K, which is $4\times$ larger.

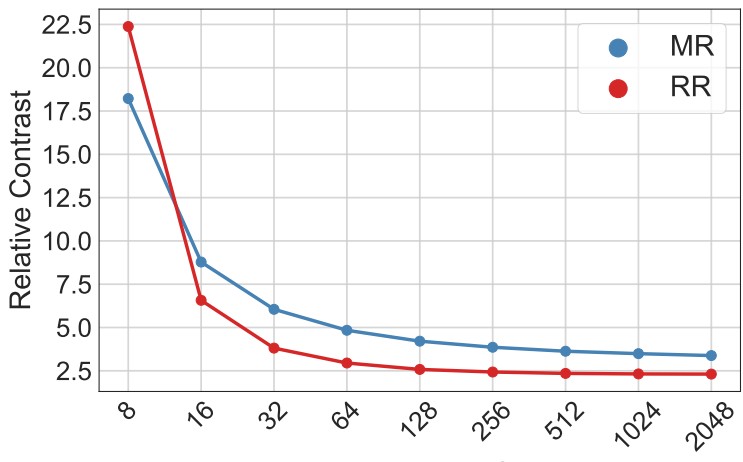

Figure 14: Relative contrast of varying capacity MRs and RRs on ImageNet-1K corroborating the findings of He et al. [18].

In this section we also examine the variation of k-Recall@N with by probing a larger search space with IVF and HNSW indices. For IVF, search probes represent the number of clusters shortlisted for linear scan during inference. For HNSW, search quality is controlled by the $efSearch$ parameter [38], which represents the closest neighbors to query $q$ at level $l_c$ of the graph and is analogous to number of search probes in IVF. As seen in Figure 13, general trends show a) an intuitive increase in recall with increasing search probes $n_p$) for fixed search probes, b) a decrease in recall with increasing search dimensionality $d$ c) similar trends in ImageNet-1K and $4\times$ larger ImageNet-4K.

## H.2    Relative Contrast

We utilize Relative Contrast [18] to capture the difficulty of nearest neighbors search with IVF-MR compared to IVF-RR. For a given database $X = \{x_i \in \mathbb{R}^d, i = 1, \ldots, N_D\}$, a query $q \in \mathbb{R}^d$, and a distance metric $D(.,.)$ we compute relative contrast $C_r$ as a measure of the difficulty in finding the 1-nearest neighbor (1-NN) for a query $q$ in database $X$ as follows:

1. Compute $D_{min}^q = \min\limits_{i=1...n} D(q, x_i)$, i.e. the distance of query $q$ to its nearest neighbor $x_{nn}^q \in X$

2. Compute $D_{mean}^q = E_x[D(q, x)]$ as the average distance of query $q$ from all database points $x \in X$

3. Relative Contrast of a given query $C_r^q = \dfrac{D_{mean}^q}{D_{min}^q}$, which is a measure of how *separable* the query's nearest neighbor $x_{nn}^q$ is from an average point in the database $x$

4. Compute an expectation over all queries for Relative Contrast over the entire database as

$$C_r = \frac{E_q[D_{mean}^q]}{E_q[D_{min}^q]}$$

It is evident that $C_r$ captures the difficulty of Nearest Neighbor Search in database $X$, as a $C_r \sim 1$ indicates that for an average query, its nearest neighbor is almost equidistant from a random point in the database. As demonstrated in Figure 14, MRs have higher $R_c$ than RR Embeddings for an Exact Search on ImageNet-1K for all $d \geq 16$. This result implies that a portion of MR's improvement over RR for 1-NN retrieval across all embedding dimensionalities $d$ [31] is due to a higher average separability of the MR 1-NN from a random database point.

## H.3    Generality across Encoders

We perform an ablation over the representation function $\phi : X \rightarrow \mathbb{R}^d$ learnt via a backbone neural network (primarily ResNet50 in this work), as detailed in Section 3. We also train MRL models [31] $\phi^{MR(d)}$ on ResNet18/34/101 [19] that are as accurate as their independently trained RR baseline models $\phi^{RR(d)}$, where $d$ is the default max representation size of each architecture. We also train

MRL with a ConvNeXt-Tiny backbone with $[d] = \{48, 96, 192, 384, 786\}$. MR-768 has a top-1 accuracy of $79.45\%$ compared to independently trained publicly available RR-768 baseline with top-1 accuracy $82.1\%$ (Code and RR model available on the official repo[6]). We note that this training had no hyperparameter tuning whatsoever, and this gap can be closed with additional model training effort. We then compare clustering the MRs via IVF-MR with $k = 2048, n_p = 1$ on ImageNet-1K to Exact-MR, which is shown in Figure 15. IVF-MR shows similar trends across backbones compared to Exact-MR, i.e. a maximum top-1 accuracy drop of $\sim 1.6\%$ for a single search probe. This suggests the clustering capabilities of MR extend beyond an inductive bias of $\phi^{MR(d)} \in$ ResNet50, though we leave a detailed exploration for future work.

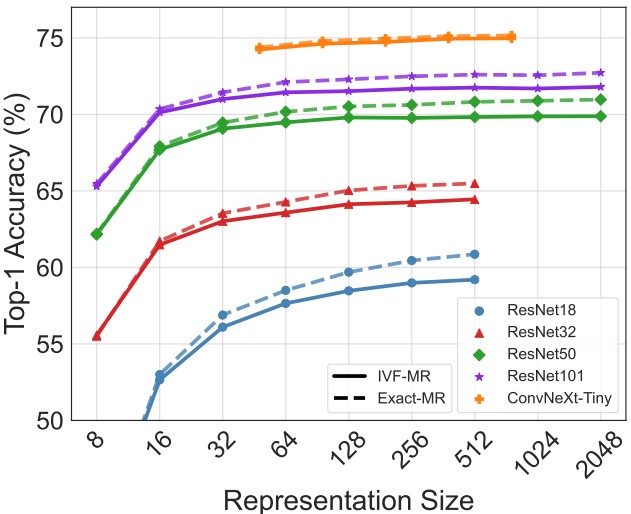

Figure 15: Top-1 Accuracy variation of IVF-MR on ImageNet-1K with different embedding representation function $\phi^{MR(d)}$ (see Section 3), where $\phi \in \{$ResNet18/34/101, ConvNeXt-Tiny$\}$. We observe similar trends between IVF-MR and Exact-MR on ResNet18/34/101 when compared to ResNet50 (Figure 9a) which is the default in all experiments in this work.

---

[6] https://github.com/facebookresearch/ConvNeXt

