# OpenReview forum: "AdANNS: A Framework for Adaptive Semantic Search"
_NeurIPS.cc/2023/Conference — NeurIPS 2023 poster_

### Official Review · Reviewer_kCbM · 2023-06-30

**Soundness:** 3 good
**Presentation:** 3 good
**Contribution:** 3 good
**Rating:** 7
**Confidence:** 4

**Summary:**

This paper proposes to use the Matryoshka Representations for approximate nearest neighbor search (ANNS). Matryoshka Representations provide the flexibility to adjust the budget for index search, index storage, and distance computation by changing the dimension of the used embedding. The paper instantiates the idea using IVF and OPQ, two popular techniques for ANNS, and shows impressive experiment results.

**Strengths:**

I like that this paper focuses on the end-to-end performance of ANNS, i.e., the top-1 accuracy of ANNS applications. This differs from the widely used recall, which only considers distance in the embedding space, and could be inspiring for the field.
1. The property of Matryoshka Representations is interesting and using it to make ANNS flexible makes sense. However, the authors could introduce more about Matryoshka Representations, e.g., how are they trained and what is the training cost compared with standard embedding.
2. Two examples of using the Matryoshka Representations are provided.
3. The experiment results are comprehensive and impressive.
4. The gains of Matryoshka Representations and limitations are discussed in Section 5.


**Weaknesses:**

The labels of the figures need to be enhanced to make them self-content.

**Questions:**

NA

**Limitations:**

Yes

---

> ### Author Rebuttal · Authors · 2023-08-06
>
> We thank the reviewer for their time and feedback and are glad they found our work impressive and well suited for ANNS. We address the reviewer’s feedback below:
>
> 1) **Details on Matryoshka Representations**: We have briefly discussed the details of Matryoshka Representations in L154 - 163 and shall add more details to make it much more accessible to readers. In our experience, Matryoshka Representation Learning does not add any training overhead or hyperparameter tuning on top of a chosen representation learning setup (across models, dataset and modalities).
> 2) **Labels of the figures**: We thank the reviewer for their feedback on figure labels. We will address these in the next revision.
>
> We hope that the rebuttal clarifies the questions raised by the reviewer. We would be very happy to discuss any further questions about the work, and would really appreciate an appropriate increase in score if reviewer's concerns are adequately addressed to facilitate acceptance of the paper.

---

> > ### Author Response · Authors · 2023-08-16
> > **Further questions or concerns?**
> >
> > We are happy to discuss if anything in the rebuttal needs more clarification or if the reviewer has further questions regarding the paper.

---

> > > ### Comment · Reviewer_kCbM · 2023-08-22
> > > **After reponse**
> > >
> > > I have read the response and decide to keep my original rating.

---

> > > > ### Author Response · Authors · 2023-08-22
> > > >
> > > > Thank you for the response and your support for the acceptance of the paper.

---

### Official Review · Reviewer_V9Qf · 2023-07-05

**Soundness:** 4 excellent
**Presentation:** 4 excellent
**Contribution:** 3 good
**Rating:** 7
**Confidence:** 5

**Summary:**

Matryoshka Representation representations have the advantage that the first m-bits of the d-dim vector can as-is serve as a good m-dim representation of the original d-dim vector. This paper demonstrates how Matryoshka Representations (MR) can be used together with approximate nearest neighbor search indices to speed up test-time inference as well as optimize for indexing cost and storage budgets for semantic search applications. The authors run extensive experiments on two datasets with up to 21 million items, carefully exploring the design space.

Update: I have updated my score to 7 after reading clarifications provided by authors.

**Strengths:**

- The paper is very well-written and easy to follow (although it requires quite a few jumps to and from the appendix to really understand the results :) ).
- The papers presents a clever use of properties of Matryoshka Representations for semantic search. Unlike standard dimensionality reduction methods like SVD or random projection, use of MR representations allows use of lower-dim representations when needed without additional computation to get low-dim representations.
- Extensive experiments and analysis of the proposed approach.

**Weaknesses:**

- Too many hyper-parameters to optimize — This might limit wide scale adoption of proposed methods.
    - A major chunk of performance boost observed comes from merely using Matryoshka Representations (MR) instead of traditional dense representations (RR). For instance, in Fig. 2, IVF built with MR gives significant improvement over IVF built with RR. While proposed adaptive IVF outperforms fixed-IVF with MR, it may require an non-trivial exhaustive search of hyper-parameters. And, as show in Fig. 10, only a small fraction of these hyper-parameter settings for adaptive-IVF-MR outperform fixed-IVF with MR representations.
    - It would be helpful to also include a practical guide of how to choose these hyper-parameters in practice. In my opinion, this can significantly help wide-scale adoption of ideas in this work.
- (Minor) The proposed adaptive search methods only work with Matryoshka Representations and can not be applied directly to any dense vector representations. This might limit the impact of the paper as the user may not have the budget or infrastructure to train MR representations especially when such embeddings are obtained form some large pre-trained vision/language models.

**Questions:**

- Is ground-truth 1-NN (as per the dense representations) used or ground-truth label (as per labelled data for downstream task) used for eval?
- Can authors elaborate on why using 2k clusters with d/2 dim vectors is just as costly as using k clusters with d-dim vectors? What cost is being referred in this statement? Is it test-time inference cost or is it indexing cost?
- In Fig.2, why does AdANNS-IVF perform better than AdANNS-IVF-D? Why does using a smaller dim MR embedding give better results than using full 2048-dim embedding for clustering?

**Limitations:**

Yes

---

> ### Author Rebuttal · Authors · 2023-08-08
>
> We thank the reviewer for their time and feedback. We are glad the reviewer found our work easy to follow and clever. Below, we answer the questions raised in the review:
>
> 1) **Jumps to Appendix**: Thanks for letting us know of requiring jumps to the appendix, we shall improve readability to minimize this in the next revision.
>
> 2) **Too many hyperparameters**: As mentioned in the general comment, forming optimal index can have a massive impact on cost, QPS, latency at industry scale. So, it is an industry practice to have a careful parameter search for all the hyperparameters in ANNS. So AdANNS doesn't fall beyond that practice.
> However, we acknowledge that choosing the optimal hyperparameters for AdANNS is an interesting and open problem that requires more rigorous examination.
> - *Existing solution*: Autofaiss [1] optimizes hyper-parameters of ANNS indices for the constraints at hand (i.e. the optimal compute-accuracy model for a given use case). We will publish a similar guide on top of AdANNS in the next revision.
> - *Hyperparameter selection guide*: We agree with the reviewer that only a small fraction of the hyperparameter space of AdANNS is above IVF-MR – Fig. 10 was intended to show the entire design search space, which is not feasible in real-world deployment. Hence, we suggest starting with IVF-MR configs and moving towards AdANNS based on the use case.
> We would like to mention that the gains of AdANNS-IVF come from both simple adaptivity (e.g. MG-IVF-RR is 1-1.5% more accurate than IVF-RR) and better clustering of MRs (IVF-MR is 1-1.5% more accurate than IVF-RR), resulting in an overall boost of up to 2% ground truth top-1 accuracy; even a bump of 0.5% is potentially significant in searching web-scale long tail data. We thank the reviewer for bringing up this interesting point, and shall add a brief discussion on selecting hyperparameters for AdANNS in the next revision.
>
> 3) **Requirement for Matryoshka Representations**: We acknowledge that a current disadvantage of full-potential AdANNS is the requirement to backfill MRs (Section 5.3, L405-410). We address this issue in two ways:
> - *Adaptivity*: As we demonstrate with the MG-IVF-RR baseline in Section 4.1 (Figure 2), where we “replicate” AdANNS with existing dense vector representations (RRs) for gains of more than 1% ground truth accuracy. This is a strong method, but is several times more expensive during both training and inference than AdANNS using MRs. E.g. MG-IVF-RR requires training 10 independent rigid models, each of which require independent inference to obtain dense vectors for ANNS. Further we can also leverage adaptivity through post-hoc compression (e.g. MG-IVF-SVD), albeit not as well as explicitly trained RRs.
> - *Inducing MR behavior through fine-tuning*: We agree with the reviewer that pretraining large models with MRL is expensive to train, such as in the case of large vision/language models. We have shown that Matryoshka behavior can be induced in pretrained models by fine-tuning the BERT-Base encoder on Natural Questions train set, which we use for all Natural Questions passage retrieval in this work. A more detailed study on fine-tuning to induce MR behavior is discussed in Appendix K.1 in [2].
>
> 4) **Evaluation metric**:  For all the main experiments, we use the ground-truth label to compute top-1 accuracy (available for ImageNet and NQ datasets).  However, we also used ground-truth NN (w/ exact search) in k-recall@N to be consistent with the literature (defined Appendix C, L570-572) in experiments (Appendix E.2, Figure 8b and Appendix I.1, Figure 12). Overall, ground-truth label based top-1 is a harder and more representative metric than the widely used exact search NN based metrics. We will make this point more clear in the next revision.
>
> 5) **Clustering and cluster selection costs**: The complexity of both training a k-means clustering with d-dim vectors and the cluster selection for a given query scale as *O(kd)*. So both the training and cluster selection costs remain the same as k-clusters with d-dim vectors for 2k-clusters and d/2-dim vectors. We shall make this more clear in the next revision of the manuscript.
>
> 6) **AdANNS-IVF-D**: We discuss this in more detail in Section 5.1 (L363-373), but to summarize, AdANNS-D is an “emergent” behavior that has not been explicitly designed for, which leverages the properties of MRs to enable elastic search during inference. AdANNS-IVF provides more flexibility by making the entire design search space available for compute-aware deployment, and thus allows to find configurations that are more accurate than AdANNS-IVF-D.
>
> 7) **Low-dim clustering**: Due to the efficient information packing of MRs, for a majority of the data, we have sufficient information in d/2 dimensions when compared to the full d dimensions. This can be seen by the high top-1 classification accuracy of low-dim MRs on ImageNet. So when a pair of high-d and low-d representations contain similar information, it is easy to converge to a better clustering using low-d vectors owing to the curse of dimensionality. As k-means scales *O(kd)* and when k remains constant, low-d representations with similar accuracy converge faster and to a better solution in practice. We are happy to further discuss this aspect.
>
> We hope that the rebuttal clarifies the questions raised by the reviewer. We would be very happy to discuss any further questions about the work, and would really appreciate an appropriate increase in score if reviewer's concerns are adequately addressed to facilitate acceptance of the paper.
>
> [1] Paltz et al., 2023 criteo/autofaiss, Github repository
>
> [2] Kusupati et al., NeurIPS 2022 Matryoshka Representation Learning

---

> > ### Comment · Reviewer_V9Qf · 2023-08-14
> > **Acknowledging author response**
> >
> > Thank you for answering my questions!
> > I have read the author response and have increased my score from `6:Weak Accept` to `7: Accept` and I would vote in favor of accepting the paper.
> > I would encourage authors to a) update the presentations so as to minimize the need to oscillate between the appendix and the main paper and b) add more details on hyper-parameter tuning, maybe even a set of default hyper-parameters that the authors would suggest using for new applications/datasets.

---

> > > ### Author Response · Authors · 2023-08-14
> > > **Thanks**
> > >
> > > We thank the reviewer for their response and support for the acceptance of the paper.
> > >
> > > We shall address both the points mentioned above in the camera ready version of the manuscript.

---

### Official Review · Reviewer_kQzU · 2023-07-08

**Soundness:** 3 good
**Presentation:** 3 good
**Contribution:** 3 good
**Rating:** 5
**Confidence:** 4

**Summary:**

The authors introduced AdANNS, a framework that effectively harnesses the flexibility of Matryoshka Representations. This approach is applied to two fundamental components of typical ANNS systems: (a) the search data structure that stores datapoints, and (b) the distance computation that maps a given query to points within the data structure. To incorporate matryoshka representations with each of these ANNS building blocks, the authors proposed AdANNS-IVF and AdANNS-OPQ.

Extensive experiments demonstrated that AdANNS achieves a state-of-the-art accuracy-compute trade-off for the two primary ANNS building blocks. Moreover, when combining AdANNS-based building blocks, superior real-world composite ANNS indices can be constructed.

Despite these findings, it should be noted that the authors conducted their research on small-scale open-source datasets only. Nevertheless, they assert that the primary contribution of this research is the proposal of AdANNS, which they claim is applicable to web-scale industrial systems.

**Strengths:**

(1) Although Matryoshka Representations (MR) already contain multi-granular representations, meticulous integration with ANNS building blocks is paramount to develop a functional method, representing the key contribution of this work.
(2) Through extensive experiments, it has been demonstrated that AdANNS achieves a state-of-the-art accuracy-compute trade-off for the two primary ANNS building blocks. Furthermore, the combination of AdANNS-based building blocks results in the creation of superior real-world composite ANNS indices.

**Weaknesses:**

(1) It may be beneficial for the authors to restructure the introduction. For instance, in the second paragraph, the authors delve into the details of IVF, explaining its two phases and why using the same high-dimensional Rigid Representation (RR) for cluster mapping and linear scanning could be sub-optimal. While I understand the authors' intent to introduce adaptive representations followed by matryoshka representations, I believe it would be more logical to first describe each component of ANNS and explain how adaptive representations could serve as a replacement for rigid representations.

(2) It's important to acknowledge that the authors conducted their research exclusively on small-scale open-source datasets. Despite this limitation, they maintain that the main contribution of their study is the introduction of AdANNS, which they assert can be applied to web-scale industrial systems.

**Questions:**

Why didn't you include a larger dataset for the experiments? Was it due to data availability or limitations in your implementation?

**Limitations:**

The authors have discussed the limitations of this work in section 5.3: To use AdANNS on a corpus, one need to back-fill the MRs of the data – a significant yet a one-time overhead.

---

> ### Author Rebuttal · Authors · 2023-08-06
>
> We thank the reviewer for their time and feedback. We are glad the reviewer acknowledged our work’s potential to provide superior real-world ANNS indices. Below, we answer the questions raised in the review:
>
> 1) **ANNS components in the Introduction**: Thanks for the valuable suggestion on restructuring the introduction. We agree that fleshing out ANNS and its fundamental components. We shall add it in the next revision.
>
> 2) **Evaluation of AdANNS**: We respectfully disagree with the reviewer that the evaluation was done on small scale datasets. While not Billion scale like in the industry, ImageNet (image, 1.3M database) and Natural Questions (text, 21M database) are strong benchmark datasets for retrieval evaluation with publicly available raw data to train new representations from scratch (which is needed for strong adaptive representations and new variants like MRs). Most of the ANNS progress was on pre-built representations with around a Million points in the database, such as in ANN-Benchmarks [1]. The more recent larger benchmarks released (Big ANN Benchmarks [2]) have about 10M database points. Both these standard benchmarks rarely have publicly available raw data – which is unlike ImageNet and NQ that are of similar scale. Further, none of the ANNS benchmarks have ground truth labels for evaluation, instead considering exact search NN as ground truth for evaluation. Because our implementations easily scale to NQ, they should work for most of these ANNS benchmarks; as shown previously [3], the nontrivial gains achieved on ImageNet-based datasets often generalize to other datasets and to web-scale.
>
>     In summary, it is primarily due to the lack of publicly available raw data for the other datasets and not the limitations in implementation. We discuss this in more detail in Related Work (L130-140) and are happy to discuss this further.
>
>
> We hope that the rebuttal clarifies the questions raised by the reviewer. We would be very happy to discuss any further questions about the work, and would really appreciate an appropriate increase in score if reviewer's concerns are adequately addressed to facilitate acceptance of the paper.
>
> [1] Aumüller et al., Information Systems 2019 ANN-Benchmarks: A Benchmarking Tool for Approximate Nearest Neighbor Algorithms
>
> [2] Simhadri et al., Big ANN Benchmarks, Github repository
>
> [3] Kornblith et al., CVPR 2019 Do Better ImageNet Models Transfer Better?

---

> > ### Author Response · Authors · 2023-08-16
> > **Further questions or concerns?**
> >
> > We are happy to discuss if anything in the rebuttal needs more clarification or if the reviewer has further questions regarding the paper.

---

> > > ### Comment · Reviewer_kQzU · 2023-08-18
> > >
> > > Thank you for your reply. Throughout the paper, the authors emphasize that the proposed framework is suitable for web-scale search systems. Consequently, the audience would anticipate that the authors present a large-scale, real web-scale experiment. If not, I'd suggest the authors reconsider the emphasis on web-scale. I will keep my score the same.

---

> > > > ### Author Response · Authors · 2023-08-18
> > > >
> > > > Thanks for the suggestion. We agree that a real web-scale (Billion-scale) experiment would drive the point home much more significantly than using the benchmark datasets (of up to 21 Million-scale).
> > > >
> > > > While we believe that AdANNS can scale reliably to web-scale, we understand the reviewer's point of view and will reconsider the emphasis on web-scale in the next revision of the paper to make it more grounded.

---

### Official Review · Reviewer_i99a · 2023-07-09

**Soundness:** 3 good
**Presentation:** 3 good
**Contribution:** 2 fair
**Rating:** 4
**Confidence:** 4

**Summary:**

The author introduced an adaptive method for searching near-neighbors called AdANNS, which employs different representations of the same item at various stages of the engine. Rather than relying on traditional fixed vector representations, the authors utilized Matryoshka representations, creating a nested representation with varying dimensions for each item.

**Strengths:**

1) AdANNS beats the traditional IVF and OPQ on accuracy and compute-resource tradeoff by proposing two main ANNS building blocks: search data structures (AdANNS-IVF) and quantization (AdANNS-OPQ).
2) The comparisons are made against the standard NN approaches IVF and OPQ.



**Weaknesses:**

1) Limited datasets: The authors have only used ImagNet-1K and NQ datasets for vector-based retrieval. Many standard open-source datasets are available for this purpose (ANN benchmark and BigANN benchmark). A simple MLP/autoencoder can be trained for varying embedding sizes, or dimensionality reduction (like PCA, random projection) methods can be used.
2) The paper is weak in novelty. It uses the existing method Matryyoshika Representations of each vector for the near neighbor retrieval. I do acknowledge that the modern composite ANNS index framework is being modified to support the Matryyoshika Representations. This provides a gain in top-1 accuracy for a given compute budget. However, I see this as a trivial change.
3) There is no comprehensive theoretical analysis that compares the AdANNS-IVF, AdANNSOPQ, and AdANNS-DiskANN over their baseline counterparts.
4) Baselines: The baselines comparison are limited. As AdANNS can be adapted to HNSW and Scann as well, it will be good to analyze the performance improvement there.
5) Marginal gains: (Table 3 Appendix) It seems like AdANNS-IVF provides marginal accuracy gains over IVF with similar query times. This is good but not very convincing in favor of Matryoshka representations against single rigid representations.

**Questions:**

1)  How are the vector sizes $d_c$ and $d_s$ decided for a particular dataset? A grid search is not practical. Similarly how the parameter $k$ is set? Is there any rule?

**Limitations:**

The authors have discussed the limitations in section 5.3. There are no potential negative societal impacts.

---

> ### Author Rebuttal · Authors · 2023-08-08
>
> We thank the reviewer for their time and feedback. Below, we answer the questions raised in the review:
>
> 1) **Limited datasets**: We discuss our reasoning to experiment on only ImageNet and Natural Questions datasets in more detail in Related Works (L130-140). To summarize:
> - *Existing ANNS benchmarks*: BigANN benchmarks [1] and ANN benchmark datasets [2] **do not provide raw data**, and we thus cannot encode MRs for AdANNS. Also they are in general datasets with <10M points. Results on ImageNet and NQ demonstrate the potential of AdANNS across modalities (image, text) and scale (1M to 21M). Finally, dense retrieval community uses NQ as benchmark and similarly, non-trivial gains on ImageNet benchmarks often generalize (Kornblith et al. [3]), even to web-scale.
> - *Training MLP/Autoencoder*: As both  BigANN [1] and ANNS benchmarks [2] do not have ground truth labels, we would need to train an MLP/autoencoder with a reconstruction loss on the embeddings, which is not ideal to learn optimal information packing of MRs. We would also require raw data to finetune a RR encoder to an MR encoder, as we do for BERT-Base on Natural Questions train set (used for all passage retrieval experiments on NQ in this work) and is also shown by Kusupati et. al [4], Appendix K.1
> - *Dimensionality reduction*: We would like to point out that PCA or Random projections do not work well in practice as shown in our experimentation (see MG-IVF-SVD in Section 4.1, Figure 2) and Table 2 in Kusupati et al. [4].
>
> 2) **Novelty**: We strongly disagree that the AdANNS framework is a trivial change over MRL. Despite vast amount of literature on ANNS going back decades, existing methods all used a fixed representation size for all building blocks. We are proposing a shift in this paradigm, where different blocks can use different parts of MR representations thus providing one more point of control in ANNS pipeline. Given the significance of ANNS in search, retrieval augmented LLMs, we believe the improved compute-accuracy trade-offs by AdANNS can have significant downstream impact.  We discuss this in more detail in the Introduction (L077-085) and are happy to discuss further.
>
> 3) **Theoretical analysis**: Thanks for the comments on potential theoretical analysis. Generally, ANNS field itself lacks significant rigorous analysis due to challenges in modeling the data and query. In particular, even famous techniques like IVF, HNSW do not have solid theoretical analysis which is mostly limited to methods like KD-trees which do not scale. Having said that, we agree that further investigation on theoretical front of ANNS is needed, and we leave it for future work.
>
>
> 4) **Baseline evaluation**: We respectfully disagree that our baseline evaluations are limited. We in fact already evaluated HNSW+AdANNS:  show results on AdANNS-HNSW with OPQ in Appendix D (Figure 6d), where we find that AdANNS provides gains over using MRs and RRs in high compression regimes (<= 32 bytes). We further evaluate AdANNS-HNSW with a recall score analysis in Appendix I (Figure 12).
> Similarly, ScANN is essentially a search space partitioning with anisotropic vector quantization, which is analogous to IVF with OPQ which we have experimented with. Lastly, we have discussed the future extension of AdANNS to HNSW and ScANN in Section 4 (L278-284) and are happy to discuss in more detail.
> We would like to highlight that both IVF and DiskANN are state-of-the-art techniques used in billion-scale data regimes, and OPQ is a ubiquitous quantization scheme for ANNS.
>
> 5) **Marginal accuracy gains**: We strongly disagree with the reviewer that the accuracy gains for AdANNS over rigid IVF are marginal.
> Note that even gains of 0.5% are potentially significant for web-scale long tail data and also translate to other real-world tasks [4]. In contrast, we are able to gain up to 1.5% ground truth accuracy over rigid representations for the same query time (Appendix B, Table 3).
>
> 6) **Hyperparameter search ($d_c$, $d_s$)**: We acknowledge that choosing the optimal hyperparameters for AdANNS is an interesting and open problem that requires more rigorous examination, though we would like to point out that tuning for constraints is not a new problem in the ANNS community [3]. We find that generally speaking, the highest dimensionality that fits within a compute budget is optimal for IVF (Figure 14, Appendix I.3; Figure 7c, Appendix E.1). For OPQ, we find that across ANNS strategies the best performance comes from learning quantization on a smaller dimensionality (Section 4.2, L310-312). Lastly, we also monitor the accuracies of low-d representations on the downstream dataset to ensure only a minimal loss in information from high-d representations. We shall add these takeaways as a guide for setting the right $d_c$ and $d_s$ for a given dataset.
>
> 7) **Number of Clusters ($k$)**: It is generally accepted that a good starting point for the number of clusters $k$ is $\sqrt{N_D/2}$ where $N_D$ is the number of indexable items [5]. And $k=\sqrt{N_D}$ is the optimal choice of $k$ from a FLOPS computation perspective as can be seen in Appendix E.4. We thank the reviewer for bringing up this point, and will discuss this in the next revision.
>
> We hope that the rebuttal clarifies the questions raised by the reviewer. We would be very happy to discuss any further questions about the work, and would really appreciate an appropriate increase in the score if the reviewer’s concerns are adequately addressed to facilitate acceptance of the paper.
>
> [1] Simhadri et al., Big ANN Benchmarks, Github repository
>
> [2] Aumüller et al., Information Systems 2019 ANN-Benchmarks: A Benchmarking Tool for Approximate Nearest Neighbor Algorithms
>
> [3] Paltz et al., 2023 criteo/autofaiss, Github repository
>
> [4] Kornblith et al., CVPR 2019 Do Better ImageNet Models Transfer Better?
>
> [5] Mardia et al., 1979 Multivariate Analysis p.365

---

> > ### Author Response · Authors · 2023-08-16
> > **Further questions or concerns?**
> >
> > We are happy to discuss if anything in the rebuttal needs more clarification or if the reviewer has further questions regarding the paper.

---

### Author Rebuttal · Authors · 2023-08-08

We thank the reviewers for their time and valuable feedback. We are happy to know that the reviewers found the paper to be very well written, easy to follow, and clever along with extensive experimentation and analysis showcasing state-of-the-art accuracy-compute tradeoff for ANNS building blocks and composite indices. However, there were some general comments we want to address here briefly before diving deep in each individual rebuttal.

1) **Further improving readability**: We shall fix the remaining few issues with the flow of the paper and try to reduce dependency on referring to external papers and the appendix for the readers when focusing on the fundamental and primary contributions.
2) **Hyperparameter search guide**: Most reviewers found a need for a hyperparameter search guide for AdANNS and we completely agree with them. While the hyperparameter search for AdANNS might seem expensive, it is not a new problem in the ANNS community which has potential solutions like Autofaiss [1]. Furthermore, because the index is formed once and is used for potentially billions of queries thus having massive implications for cost, latency and query-per-second, generally hyperparameter search for the best index is an acceptable industry practice.
In case of AdANNS, starting at the best configurations of MRs followed by a local design space search would lead to near-optimal AdANNS configurations (e.g. use IVF-MR to bootstrap AdANNS-IVF). We shall add a section on this in the next revision of the paper.
3) **Limited datasets**: Another common question is regarding the evaluation of AdANNS on only ImageNet (1M and 4M) and Natural Questions (21M) datasets. We discuss this in detail as part of our Related Work (L130-140), but a short answer is that all the standard ANNS benchmarks [2,3] *do not provide raw data* to allow for the training of new representations. This makes evaluation near impossible with new representation learning paradigms. At the same time, using post-hoc compression techniques to obtain adaptive representations results in extremely poor performance as shown in this paper and MRL [4]. Lastly, even the benchmark datasets are often in the scale of 1-10M and are similar to what we have evaluated – note that ImageNet and NQ have ground truth labels allowing us to compute the hard and more representative ground truth top-1 accuracy, unlike most benchmarks. We are happy to discuss in more detail during the discussion phase.
4) **Novelty and baselines**: While most reviewers agree that enabling adaptive semantic search in ANNS is a hard problem, we want to re-emphasize that without meticulous integration of adaptive representations into ANNS building blocks and composite indices, the utility of adaptive representations is severely limited. While MRL [4] proposed a simple variant of adaptive retrieval, in practice it does not scale to any large-scale problems due to various systems issues often already addressed in modern-day ANNS pipelines. We believe that AdANNS is a non-trivial contribution that shows the utility of adaptive representations across core building blocks (that form the baselines) along with a scalable implementation.

We hope that the individual rebuttals clarify the questions raised by the reviewers. We are happy to discuss any other concerns further during the discussion phase and appreciate your support for the acceptance of the paper.

[1] Simhadri et al., Big ANN Benchmarks, Github repository

[2] Aumüller et al., Information Systems 2019 ANN-Benchmarks: A Benchmarking Tool for Approximate Nearest Neighbor Algorithms

[3] Paltz et al., 2023 criteo/autofaiss, Github repository

[4] Kusupati et al., NeurIPS 2022 Matryoshka Representation Learning

---

### Decision · Program_Chairs · 2023-09-21

**Decision:**

Accept (poster)

**Comment:**

The paper explores the use of Matryoshka Representations in approximate nearest neighbor search indices to enhance test-time inference speed, indexing cost, and storage budgets.

All reviewers agree that AdANNS demonstrates strong performance on both accuracy and compute-resource trade-offs in comparison to traditional methods like IVF and OPQ. The system is found to have state-of-the-art accuracy-compute trade-offs for the two main ANNS building blocks (kQzU, V9Qf). The paper's unique application of Matryoshka Representations is highly regarded (kQzU, V9Qf, kCbM). Reviewers appreciate how MRs offer flexibility and semantic richness to the ANNS system.
Reviewer kCbM specifically mentions that the paper brings a fresh perspective by focusing on the end-to-end performance of ANNS, which diverges from the traditional focus on recall metrics.
Overall, the reviewers found the paper to be strong in its novel approach, extensive validation, and its shift of focus towards a more holistic evaluation of ANNS systems.

Some common weaknesses raised by reviewers include: The paper's evaluations are constrained to a limited set of datasets like ImageNet-1K and NQ, missing out on other standard benchmarks (i99a, kQzU). Baseline comparisons are also considered insufficient (i99a). V9Qf highlights that the many hyper-parameters required for optimization could impede wide adoption. The system's dependency on Matryoshka Representations also limits its broader applicability.